# The Rab6-regulated KIF1C kinesin motor domain contributes to Golgi organization

Peter L Lee, Maikke B Ohlson[†], Suzanne R Pfeffer*

Department of Biochemistry, Stanford University School of Medicine, Stanford, United States

**Abstract** Most kinesins transport cargoes bound to their C-termini and use N-terminal motor domains to move along microtubules. We report here a novel function for KIF1C: it transports Rab6A-vesicles and can influence Golgi complex organization. These activities correlate with KIF1C's capacity to bind the Golgi protein Rab6A directly, both via its motor domain and C-terminus. Rab6A binding to the motor domain inhibits microtubule interaction in vitro and in cells, decreasing the amount of motile KIF1C. KIF1C depletion slows protein delivery to the cell surface, interferes with vesicle motility, and triggers Golgi fragmentation. KIF1C can protect Golgi membranes from fragmentation in cells lacking an intact microtubule network. Rescue of fragmentation requires sequences that enable KIF1C to bind Rab6A at both ends, but not KIF1C motor function. Rab6A binding to KIF1C's motor domain represents an entirely new mode of regulation for a kinesin motor, and likely has important consequences for KIF1C's cellular functions.

*For correspondence: pfeffer@stanford.edu

Present address: †Genentech, South San Francisco, United States

## Introduction

Kinesin superfamily proteins (KIFs) are microtubule-based motors that are responsible for the motility of membrane-bound compartments and transport vesicles (*Hirokawa et al., 2009b*; *Verhey and Hammond, 2009*). Of fundamental interest is how these motor proteins link to specific membrane cargoes and how they are regulated. Rab GTPases represent a family of more than 60 human proteins that mark distinct membrane-bound compartments and function in transport vesicle formation, motility, docking, and fusion (*Stenmark, 2009*; *Hutagalung and Novick, 2011*). Rabs help connect motors to their cargoes, usually via an intermediary linking protein. For example, the Rab27 Slac2 effectors recruit myosin Va (reviewed by *Fukuda, 2013*), Rab3 effector, DENN/MADD links KIF1β and KIF1A to Rab3 on synaptic vesicles (*Niwa et al., 2008*) and both Rab6 and Rab7 interact with cytoplasmic dynein via the dynactin complex (*Short et al., 2002*), bicaudal-D (*Matanis et al., 2002*), or RILP (*Jordens et al., 2001*) proteins. KIF5B also links to Rab6-containing membranes via the Rab6 effector, bicaudal-D2 (*Grigoriev et al., 2007*). Rab6 binds to myosin II (*Miserey-Lenkei et al., 2010*) and Rab5 GTPase participates indirectly in the recruitment of the plus-end directed kinesin, KIF16B to early endosomes (*Nielsen et al., 1999*; *Hoepfner et al., 2005*).

KIF1C is a member of the Kinesin-3 family that includes the Unc104/KIF1A motor that transports synaptic vesicles to growth cones (*Hirokawa et al., 2009b*). KIF1C has been reported to be a Golgi-localized, tyrosine phosphorylated protein that interacts with the protein tyrosine phosphatase PTPD1 (*Dorner et al., 1998*) and bicaudal-D-related protein 1 (BICDR-1) (*Schlager et al., 2010*). Phosphorylation of a carboxy-terminal serine allows binding to 14-3-3 proteins (*Dorner et al., 1999*). KIF1C was initially reported to participate in the transport of proteins from the Golgi to the endoplasmic reticulum (ER; *Dorner et al., 1998*), but subsequent gene disruption in mice yielded animals with no apparent abnormalities, and fibroblasts from these mice showed normal Golgi to ER

**eLife digest** Within our cells there are many compartments that play important roles. Small bubble-like packages called vesicles carry proteins and other molecules between these compartments. These vesicles can be driven around cells by a family of motor proteins called kinesins, which move along a network of filaments called microtubules.

Kinesin proteins have two sections known as the N-terminus and the C-terminus. In most cases, the N-terminus contains the motor that binds to and walks along microtubules, while the C-terminus binds to vesicles or other cell compartments. Attached to the compartments are members of another family of proteins called the Rab GTPases. These proteins help the kinesins bind to a compartment, but it was not clear if, or how, these proteins control the activity of the kinesins.

Here, Lee et al. studied a kinesin called KIF1C. The experiments show that this kinesin can move vesicles that contain a Rab-GTPase called Rab6A along microtubules. Unexpectedly, Rab6A controls the activity of KIF1C by directly interacting with the motor as well as the C-terminus. Loss of the kinesin from the cell slows down the delivery of cargo carried in vesicles to the surface of the cell.

The experiments also show that KIF1C is involved in organizing another compartment within cells called the Golgi. This role relies on Rab6A binding to both the N-terminus and C-terminus of the kinesin, but does not require the kinesin to act as a motor. Lee et al.'s findings reveal a new way in which the activity of kinesins can be controlled. Future challenges will be to find out if other kinesins are also controlled in this way and discover when and where the Rab GTPases bind motor domains in cells.

transport (*Nakajima et al., 2002*). More recent studies have shown that KIF1C acts to regulate podosome dynamics in macrophages (*Kopp et al., 2006*; *Efimova et al., 2014*; *Bhuwania et al., 2014*) and is also important in vesicle transport in neurons (*Schlager et al., 2010*), MHC class II antigen presentation in myeloid cells (*del Rio et al., 2012*), and α5β1-integrin transport (*Theisen et al., 2012*). Consistent with these findings, KIF1C was identified in a genome-wide screen for proteins important for VSV-G transport to the cell surface; its depletion also led to fragmentation of the Golgi ribbon as monitored by GM130 localization (*Simpson et al., 2012*).

We show here that Rab6A regulates the function of KIF1C by direct interaction with both KIF1C's C-terminal cargo binding domain and, more surprisingly, with its N-terminal motor domain. Rab6A binding to the motor domain blocks KIF1C interaction with microtubules and inhibits KIF1C motility. We confirm that depletion of KIF1C leads to fragmentation of the Golgi and show that while both N- and C-terminal Rab6A binding sites are required for KIF1C rescue, KIF1C motor activity is not: neither ATP nor microtubule binding is required. Finally, Rab6A-decorated vesicles are less confined and less directed in cells depleted of KIF1C, consistent with its role as a Rab-regulated motor that aids in intra- and post-Golgi vesicle transport. These data reveal a novel form of motor regulation with unexpected consequences for motor function in Golgi organization.

## Results

### KIF1C binds to Rab6A GTPase

KIF1C is comprised of an N-terminal motor domain that is highly homologous to KIF1A and KIF1B, followed by several coiled coil stretches that are interrupted by a Forkhead homology domain (FHA; *Figure 1A*). KIF1C was identified as a protein tyrosine phosphatase D1 (PTPD1) binding partner whose binding domain is located between the third and fourth coiled coil (*Dorner et al., 1998*). The KIF1C C-terminal domain binds both 14-3-3 proteins (*Dorner et al., 1999*) and the Rab6A effector, bicaudal-D-related protein 1 (*Schlager et al., 2010*).

We identified KIF1C in a two hybrid screen for Rab GTPase binding partners (*Reddy et al., 2006*). More detailed studies showed that transiently expressed, full-length myc-KIF1C and GFP-Rab6A could be co-immunoprecipitated (*Figure 1B*). However, this interaction may have been indirect as BICDR-1 is both an effector of Rab6A and a KIF1C binding partner (*Schlager et al., 2010*). Direct and specific binding between Rab6A and KIF1C was tested using purified GST-constructs. As shown in *Figure 1C*, KIF1C's C-terminal 40 amino acids, termed the C-terminal binding domain

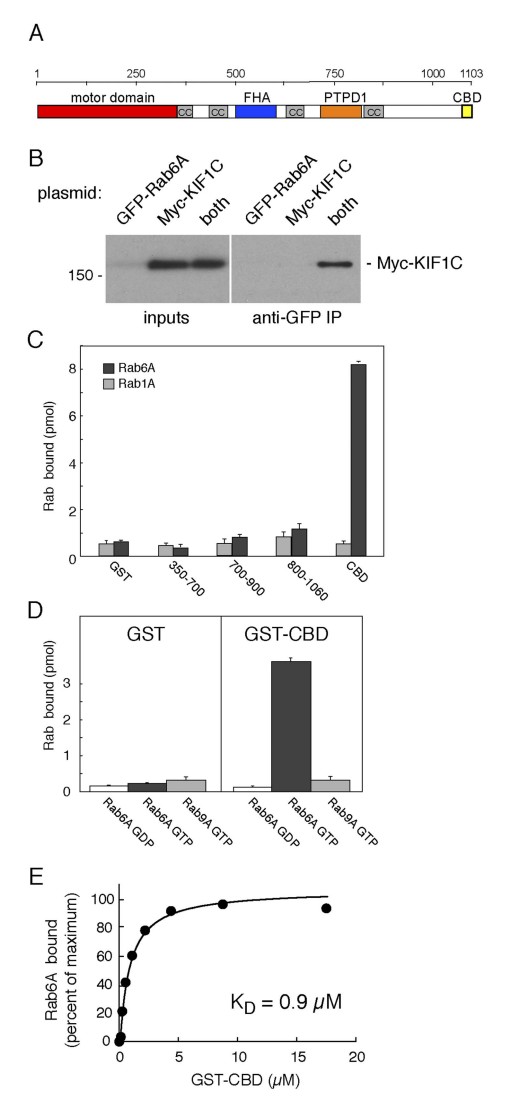

**Figure 1**. KIF1C binds Rab6A GTPase. (**A**) KIF1C schematic showing the N-terminal motor (red), predicted coiled-coil (cc, gray), Forkhead homology (FHA, blue), protein tyrosine phosphatase D1 binding (PTPD1, orange), and C-terminal Rab binding (CBD, yellow) domains (AA 1060-1103). (**B**) KIF1C co-immunoprecipitation by Rab6A. HEK293 cells were transfected with GFP-Rab6A, myc-KIF1C, or both. Lysates were immunoprecipitated with llama GFP-binding protein and immunoblotted with anti-myc antibody. Left panel, total soluble cell lysates, 5%; right panel, bound fraction, 33%. (**C**) Domain specificity of Rab6A binding. His-Rab6A-[35]S-GTPγS (black bars) or His-Rab1A-[35]S-GTPγS (gray bars) (1 μM) binding to GST-KIF1C constructs (15 μM) pulled-down with glutathione Sepharose and quantified by liquid scintillation counting (error bars = SD). (**D**) Nucleotide specificity of Rab6A binding. Binding of His-Rab6A or His-Rab9A (500 nM) preloaded with [35]S-GTPγS or [3]H-GDP to GST or GST-KIF1C CBD (15 μM) was assayed as in **C**

(CBD; *Figure 1A*), bound to purified His-Rab6A protein (*Figure 1C–E*). Binding to His-Rab6A by N-terminally truncated KIF1C was observed only with constructs containing the CBD (*Figure 1C*). Importantly, binding of the CBD required that His-Rab6A be in its active, GTP-bound state, as would be expected for a Rab effector (*Figure 1D*); binding was also Rab6A-specific as neither His-Rab1A (*Figure 1C*) nor His-Rab9A (*Figure 1D*) bound this domain strongly. His-Rab6A binding to the CBD was saturable and concentration dependent with a $K_D$ of 0.9 μM, consistent with the affinity of most Rab protein:effector interactions (*Burguete et al., 2008*).

To verify whether the CBD was the only site of Rab6A interaction, we tested Rab6A binding to KIF1C lacking the CBD (*Figure 2A*, KIF1C ΔCBD), synthesized by in vitro transcription and translation. (Significant amounts of active full-length protein could not be obtained upon expression in bacteria, consistent with work from Hirokawa on KIF1A motor protein [*Nitta et al., 2004*].) Surprisingly, deletion of the CBD in the context of full-length KIF1C did not abolish binding to GST-Rab6A (*Figure 2B*). Similarly, KIF1C constructs containing the first 500, 400 (data not shown), or 350 amino acid residues comprising only the motor domain (KIF1C-350) bound specifically to GST-Rab6A (*Figure 2C*, left). In contrast, a KIF1C construct lacking both the motor domain and the CBD (Δmotor ΔCBD) failed to bind GST-Rab6A (*Figure 2C* right). KIF1C motor domain binding to Rab6A was specific as no significant binding was seen with either GST-Rab9A or GST-Rab5A proteins (*Figure 2D*). In concentration-dependent binding analyses, Rab6A was capable of binding up to 25% of in vitro translated KIF1C motor domain, suggesting only a quarter of the molecules synthesized by in vitro translation were active. These experiments yielded an apparent $K_D = 0.23$ μM (*Figure 2E*), strong for a Rab: effector interaction. These data show that Rab6A binds specifically to KIF1C at both its N- and C-termini.

## Rab6A binds directly to the KIF1C motor domain

Because of the novelty of a Rab GTPase–motor domain interaction, it was essential to determine if Rab6A binding to the motor domain is direct. For this purpose, we took advantage of the strategy employed by Hirokawa and coworkers in their studies of KIF1A to create a stable KIF1C motor domain (*Nitta et al., 2004*).

*Figure 1. Continued*

(error bars = SD). (**E**) His-Rab6A Q72L binds to GST-KIF1C CBD in a concentration-dependent manner. GTPγS-loaded His-Rab6A Q72L (0.58 µM) binding to GST-KIF1C CBD (immobilized on glutathione Sepharose) as determined by quantitative fluorescent antibody immunoblot, presented as a fraction of maximal binding (2.2% of total). Data were fit using GraphPad Prism software.

Specifically, we expressed in bacteria a construct comprised of KIF1C residues 1–349 followed by 6 residues (329–334) of KIF5C heavy chain (plus 6 His residues; 'KIF1C-355-His'). This KIF1C motor domain was purified by Ni-NTA-chromatography followed by ion exchange chromatography on Q Sepharose FF (*Figure 3A*). This method yielded a single, ~40 kD polypeptide upon Coomassie-stained SDS-PAGE.

Binding of this recombinant KIF1C-355-His motor domain to Rab GTPases was tested using GST-tagged Rab proteins bound to glutathione resin. Quantitative analysis of Rab6A binding to the *E. coli*-produced motor domain showed that binding was of high affinity (*Figure 3B*), essentially identical with that measured for KIF1C motor domain produced by in vitro translation (*Figure 2*). As shown in *Figure 3C*, recombinant KIF1C motor domain bound directly to GST-Rab6A but not to GST-Rab5A. Moreover, binding was reduced with Rab6A protein lacking its C-terminal hypervariable domain. We have shown previously that hypervariable domains are important for the binding of many Rab effectors (*Aivazian et al., 2006*; *Burguete et al., 2008*).

GST-Rab6A exchanged bound GTP less readily than His-Rab6A, making it difficult to compare His-tagged motor binding to Rab6A in GDP vs GTP without losing overall Rab6A activity. Instead, we compared motor domain binding for the active, Rab6A hydrolysis-deficient mutant (GST-Rab6A Q72L) vs the inactive, GDP-preferring form (GST-Rab6A T27N) and found strong preference for the active Rab6A mutant (*Figure 3D*). We also tested Rab discrimination of the motor's nucleotide state: GST-Rab6A showed strong preference for the KIF1C motor domain with AMP-PNP bound, compared with ADP (*Figure 3E*). Thus, Rab6A binds with preference to KIF1C's strong microtubule-binding state. Crystal structures of the KIF1A motor domain with either AMP-PNP or ADP bound indicate a change in the projection angle of loop 10 (*Kikkawa and Hirokawa, 2006*), which as described below, may explain the strong preference of Rab6A for AMP-PNP bound-KIF1C. These data demonstrate that active Rab6A can bind the KIF1C motor domain strongly and directly, without adaptor proteins, with preference for KIF1C in its strong microtubule-binding state.

## Motor domain loops 6 and 10 are required for Rab6A binding

KIF1A and KIF1C motor domains are 81% identical (*Figure 4—figure supplement 1B*). When the KIF1C sequence is superimposed onto the structure of KIF1A bound to tubulin (PDB, 2HXH, *Figure 4—figure supplement 1A*, *Kikkawa and Hirokawa, 2006*), the sequence differences localize primarily to loops 2, 3, 6, and 10, which are positioned away from the microtubule-binding interface (*Figure 4—figure supplement 1A*). To facilitate identification of KIF1C sequences needed for Rab6A binding, we tested whether the KIF1A motor domain binds to Rab6A. *Figure 4A,C* show that Rab6A bound to the KIF1A motor domain much more weakly (if at all), compared with KIF1C. We thus created motor domain chimeras containing N-terminal portions of the KIF1A motor domain fused to the C terminus of the KIF1C motor domain (KIF1A/1C) as well as the reverse chimera, KIF1C/1A, to narrow down the Rab6A interaction site (*Figure 4B*, *Figure 4—figure supplement 1B*). The C-terminal portion of KIF1C restored binding to Rab6A in the KIF1A/1C chimera, however, the KIF1C/1A chimera showed diminished binding (*Figure 4A*).

We next generated KIF1C chimeras containing just the divergent loop sequences derived from KIF1A. Consistent with the Rab6A binding capacity of the KIF1C/1A chimera, conversion of KIF1C's loops 6 and 10 into KIF1A sequences yielded a 'loop mutant' protein that showed significantly reduced Rab6A binding (*Figure 4C,D*). In contrast, conversion of loops 2 and 3, in the N-terminal portion, was of lower consequence (*Figure 4C,D*). Mutation of either loop 6 or 10 alone did not completely abolish binding, suggesting that both regions form contacts with Rab6A (data not shown). These regions are spatially adjacent in the KIF1A crystal structure and are likely also similarly situated in KIF1C (*Figure 4D*). Importantly, the KIF1C motor domain loop 6/10 mutant that did not bind Rab6A retained its ability to bind microtubules (*Figure 4—figure supplement 2*), indicative of proper protein folding. Furthermore, microtubule localization was not just from alternate kinesin 3

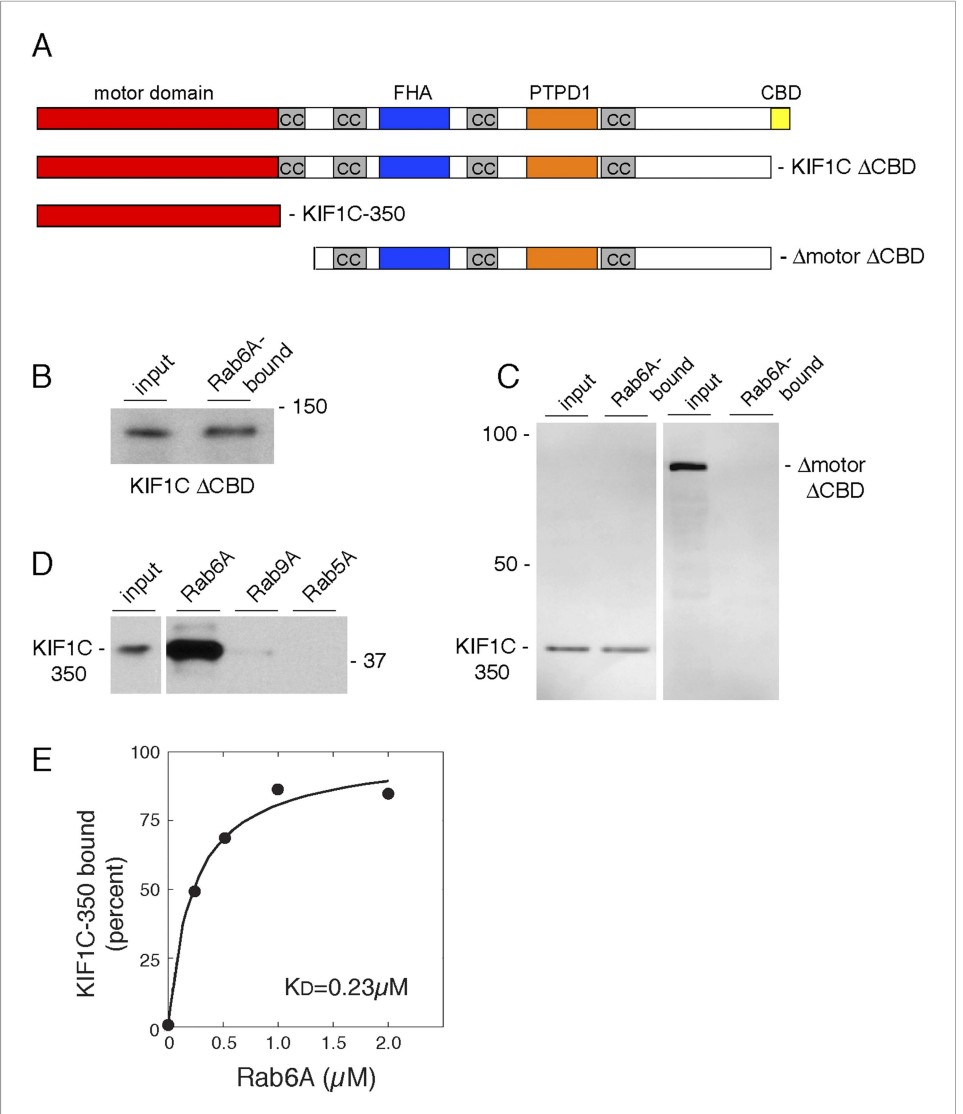

**Figure 2**. Rab6A binds KIF1C at two locations. (**A**) KIF1C schematic showing CBD truncation (ΔCBD, AA 1-1060), N-terminal motor domain (1–350), and a construct lacking both motor domain and CBD (ΔmotorΔCBD, AA 450-1060). (**B**) Binding of in vitro translated myc-KIF1CΔCBD to GTPγS-loaded GST-Rab6A Q72L (5 µM) and pulled down using glutathione Sepharose. Left, input (1%) compared to bound fraction (50%), right. (**C**) Binding of in vitro translated myc-KIF1C constructs to GTPγS-loaded GST-Rab6A Q72L (0.2 µM). Input (7%), left, compared to bound (50%), right. (**D**) Rab specificity of motor domain binding. GTPγS-preloaded GST-Rab6A Q72L, Rab9A, and Rab5A Q79L (5 µM) incubated with in vitro translated myc-KIF1C motor domain. Input, 1%, on left compared to bound fraction (48%) on right. (**E**) myc-KIF1C-350 binds to GST-Rab6A Q72L in a concentration-dependent manner. In vitro translated myc-KIF1C motor domain binding to GTPγS-loaded GST-Rab6A, presented as a fraction of maximal binding detected (25% of total). Data were fit using KaleidaGraph software. In vitro translation optimally yields a reaction concentration of 11.25 nM product.

family 'K-loop' binding, as KIF1C loop 6/10 mutant 350 with further mutations in the K-loop (NRSK or GTKT) were still able to localize to microtubules (*Figure 4—figure supplement 2*).

These in vitro binding results were confirmed by expression of KIF1C wild-type or loop mutant motor domain proteins in HEK293 cells. The wild-type motor domain was able to co-immunoprecipitate endogenous Rab6A protein, whereas similar expression of the KIF1C loop 6/10 motor domain mutant did not (*Figure 4E*). These data show that Rab6A requires loops 6 and 10 to interact with the KIF1C motor domain.

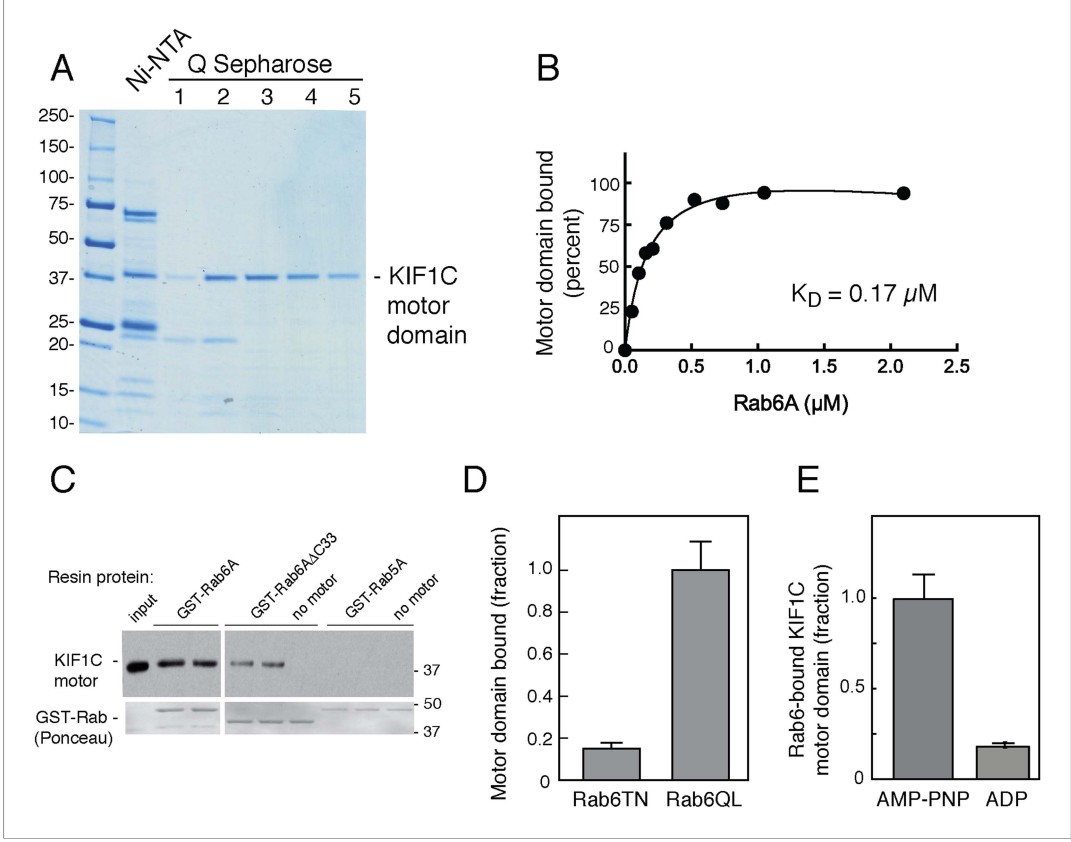

**Figure 3**. Rab6A binds directly to the KIF1C motor domain. (**A**) Coomassie-stained SDS-PAGE of bacterially expressed, KIF1C-355-His motor domain purification using Ni-NTA followed by Q Sepharose FF. Lane 1, flowthrough; lane 2, wash; lane 3–5, NaCl gradient elution. Protein from lanes 3 and 4 were used in subsequent experiments. (**B**) KIF1C-355-His binds to GST-Rab6A Q72L in a concentration-dependent manner. Purified KIF1C-355-His motor domain (*Figure 3A*, lane 4; 78.8 nM) binding to GTPγS-loaded GST-Rab6A Q72L (immobilized on glutathione Sepharose) as determined by quantitative fluorescent antibody immunoblot, presented as a fraction of maximal binding detected (61% of total). Data were fit using GraphPad Prism. (**C**) Immunoblot determination of binding of purified KIF1C-355-His (0.56 μM) to GTPγS-loaded GST-tagged QL mutant Rabs (0.2 μM) after collection on glutathione Sepharose. Input, 20% of sample; bound fraction, 37.3% of sample. Bottom panels show Ponceau S-staining to detect glutathione resin-bound and eluted Rabs. (**D**) Binding specificity of GST-Rab6A mutants T27N, GDP-preferring, and Q72L, GTP-hydrolysis deficient (0.25 μM) to purified KIF1C-355-His (45 nM) quantified as in **B**, as a percentage of maximal binding (69.1% of total). (**E**) Nucleotide binding specificity of GST-Rab6A Q72L (2.5 μM) to purified KIF1C-355-His (160 nM) in the presence of AMP-PNP or ADP as quantified by fluorescent antibody immunoblot as a percentage of maximal binding (37% of total). Error bars represent SD. Mobility of marker proteins is shown in $K_D$.

Structure prediction of the KIF1C motor domain was accomplished using PHYRE2 by taking advantage of the known KIF1A crystal structure (PDB 2ZFI, 1.55 Å resolution, PHYRE2 confidence = 100%) (*Figure 5D*; *Nitta et al., 2008*; *Kelley and Sternberg, 2009*). It is noteworthy that sequences surrounding and contributing to loop 10 are predicted to be more structured in the KIF1C model (*Figure 4D*) than in the KIF1A structure (*Figure 4B*).

In silico molecular docking was performed using ClusPro 2.0 to dock the predicted structure onto Rab6A (PDB 2Y8E) (*Kozakov et al., 2010*; *Walden et al., 2011*). Consistent with our binding studies, the top clustered model showed interaction interfaces between Rab6A switch I and II regions and loops 6 and 10 of KIF1C (*Figure 4F*). Rab6A was also docked onto KIF1A and KIF1C loop 6/10 mutant mapped onto KIF1A. Notably, while the docking confirmation of wild-type KIF1C and Rab6A involving loops 6 and 10 was highly enriched (125% increase over the next conformation), the most populated confirmations between Rab6A and KIF1A or KIF1C loop 6/10 mutant was only enriched 2% and 12%,

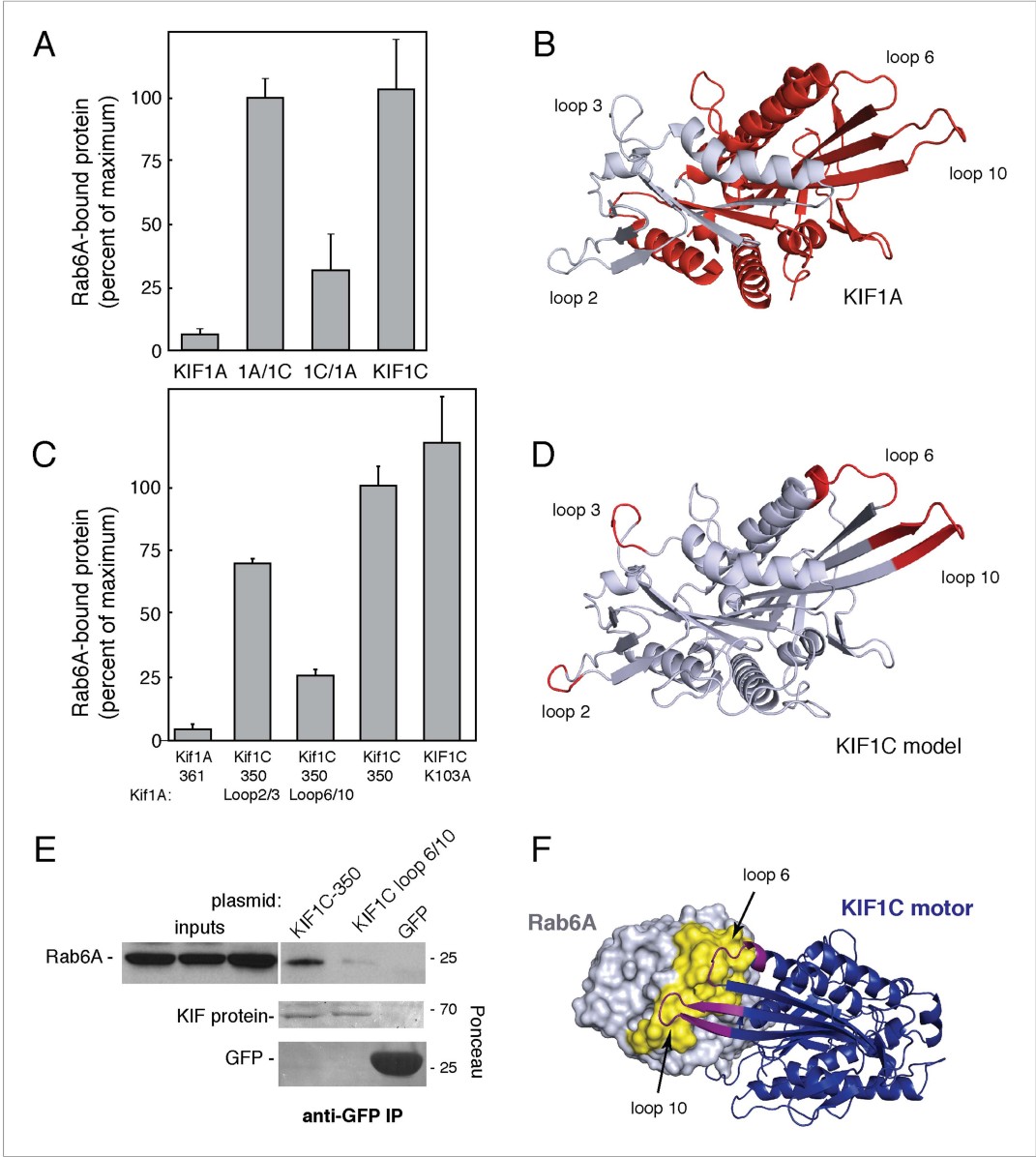

**Figure 4**. KIF1C loops 6 and 10 are necessary for Rab6A binding. (**A**) Binding of in vitro translated myc-KIF1A (1–361), myc-KIF1A/1C, myc-KIF1C/1A, and myc-KIF1C-350 motor domain chimera constructs to GTPγS-loaded GST-Rab6A Q72L (0.2 μM) presented as a percentage of maximal KIF1C bound (24.7% of input). (**B**) Crystal structure of KIF1A showing the regions swapped between KIF1A and KIF1C. The C-terminal portion is colored red. (**C**) Binding of in vitro translated myc-KIF1A (1–361), myc-KIF1C-350 Loop 2/3 swap, myc-KIF1C-350 Loop 6/10 swap, myc-KIF1C-350, and myc-KIF1C-350 K103A constructs to GTPγS-loaded GST-Rab6A Q72L (0.2 μM) presented as a percent of myc-KIF1C-350 bound (35.3% of input). (**D**) Predicted crystal structure of KIF1C with Loop 2/3 (at left), 6/10 (at right) labeled in red. The KIF1C sequence was overlaid onto the KIF1A crystal structure (PDB 2ZFI) using PHYRE2. (**E**) Binding of endogenous Rab6A to transiently expressed CFP-KIF1C-350, CFP-KIF1C-350 Loop 6/10 swap, and GFP immunoprecipitated with llama GFP-binding protein, as determined by anti-Rab6A immunoblot. Total endogenous Rab6A (2%) at left compared to bound (33%) at right. Below, bound CFP-KIF1C and GFP as measured by Ponceau S staining. (**F**) Molecular docking of Rab6A onto the predicted structure of KIF1C. The predicted structure (blue) was docked to the crystal structure of Rab6A (PDB 2Y8E, gray) using ClusPro2. The model for the largest cluster containing 144 members is shown. The switch regions of Rab6 are labeled in yellow and loop 6 and 10 of KIF1C are labeled in purple. Error bars represent SD.

The following figure supplements are available for figure 4:

**Figure supplement 1**. The motor domains of KIF1A and KIF1C share 81% identity.

*Figure 4. continued on next page*

*Figure 4. Continued*

**Figure supplement 2**. Mutations in the KIF1C K-loop do not affect KIF1C loop 6/10 mutant motor domain microtubule localization.

respectively, over their next conformations. While conformation cluster sizes are not definitive, they lend credence to the binding between KIF1C and Rab6A, as well as the importance of loops 6 and 10.

## Rab6A blocks KIF1C microtubule binding

The most obvious explanation for Rab6A binding to the KIF1C motor domain would be to influence KIF1C's microtubule binding and/or motility properties. To test if Rab6A interferes with KIF1C motor domain binding to microtubules, in vitro transcribed and translated, radiolabelled, full-length KIF1C was tested for microtubule binding in the presence and absence of purified Rab6A using a microtubule co-sedimentation assay. As a close family member to KIF1A, full-length KIF1C is likely auto-inhibited and is predicted to interact less stably with microtubules (*Hammond et al., 2009*). Nevertheless, in the presence of His-Rab6A (but not His-Rab33B), the amount of full-length KIF1C bound to microtubules decreased by more than twofold (*Figure 5A*). Especially dramatic was the ability of His-Rab6A to inhibit binding of the purified KIF1C-355-His motor domain to microtubules: KIF1C motor domain binding to microtubules could be abolished in the presence of increasing concentrations of active His-Rab6A (*Figure 5B*). Furthermore, His-Rab6A influenced both the AMP-PNP and ADP bound states of KIF1C-355-His; the presence of His-Rab6A increased the $K_D$ of KIF1C to microtubules by more than 10 fold for both AMP-PNP and ADP states (*Figure 5C,D*). The microtubule affinities observed in the absence of Rab6A (0.46 µM and 2.1 µM in AMP-PNP and ADP, respectively) are also very similar to those reported for KIF1A protein (*Soppina and Verhey, 2014*). Importantly, the specificity of this effect was confirmed by the finding that His-Rab33 did not influence the KIF1C-microtubule interaction significantly (*Figure 5D*). These data demonstrate a new mode of regulation: Rab6A can regulate the binding of KIF1C to microtubules by direct interaction with the KIF1C motor domain.

Because Rab6A binds to loop regions of the KIF1C motor domain that face away from the microtubule binding face, inhibition of microtubule binding is likely to be due to a conformational change in KIF1C, rather than direct steric hindrance. Indeed, superposition of the predicted KIF1C structure over the predicted structure of a KIF1C-Rab6A complex indicated changes in regions important for microtubule binding (not shown; site one of *Kikkawa and Hirokawa, 2006*).

To better understand the molecular mechanism by which Rab6A may regulate KIF1C, we used TIRF microscopy to analyze Rab6A's influence on the attachment of purified full-length KIF1C to microtubules. Again, as KIF1A is auto-inhibited, we expected that the majority of KIF1C would not be bound to microtubules and/or would not be fully processive (*Hammond et al., 2009*). KIF1C molecules were monitored using fluorescently labeled antibody (*Video 1*). Example frames from a video of KIF1C motility are shown in the absence (control) or presence of Rab6A protein (*Figure 5—figure supplement 1A*). The identified motors (green) that overlaid microtubules (red) are highlighted using automated segmentation (top row).

Immediately apparent was the decrease in microtubule-associated KIF1C molecules in reactions containing Rab6A. Automated segmentation (Spot Detector, *Olivo-Marin, 2002*) and tracking (u-track, *Jaqaman et al., 2008*) were used to quantify motility to support this conclusion. In a representative experiment, over 14,000 motors were quantified for each condition. Consistent with observations obtained from microtubule co-sedimentation, the presence of Rab6A resulted in a 30% decrease in the mean number of motors detected along microtubules: from 0.321 to 0.225 motors/µm ($p < 0.0001$) (*Figure 5—figure supplement 1B*). The percentage of moving motors also decreased 60% in the presence of Rab6A (from 2.13% to 0.86%; $p < 0.0001$; *Figure 5—figure supplement 1C*). Analysis of the population of moving motors yielded no significant differences in the speed or processivity of those motors in the absence of presence of Rab6A (distribution of motors overlapped for both metrics, data not shown) and likely Rab6A only affects microtubule attachment of KIF1C and no other characteristics. In summary, the presence of Rab6A led to both a reduction in the number of motors present on microtubules and a reduction in the percentage of motors that were moving.

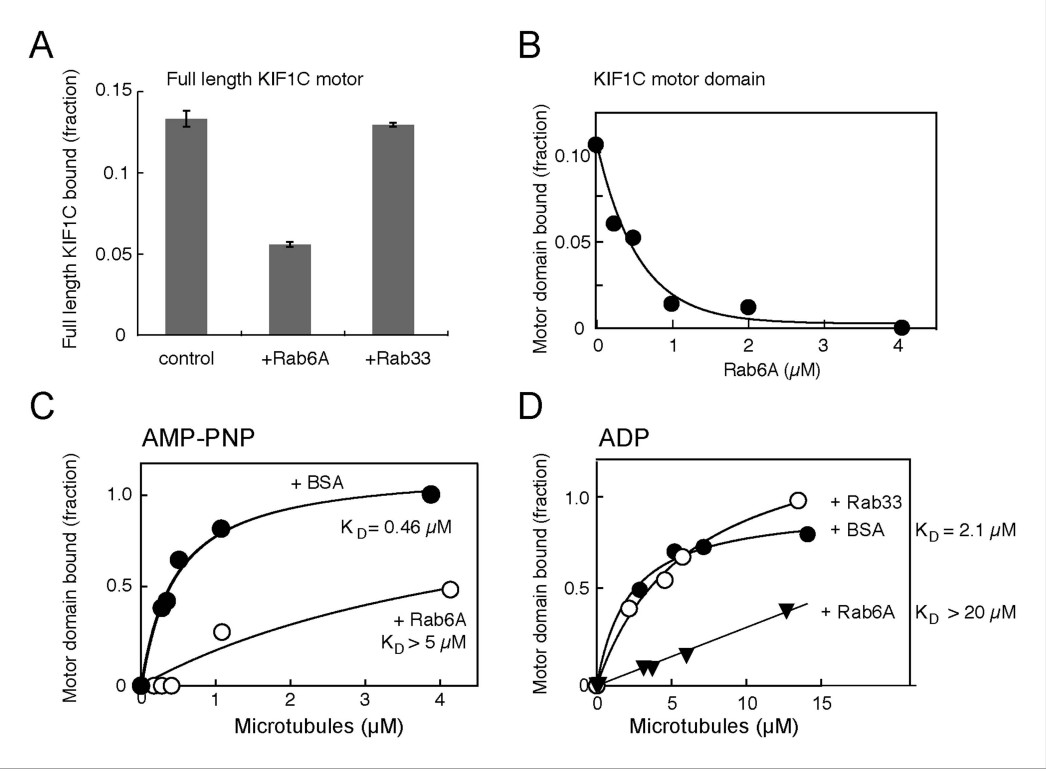

**Figure 5**. Rab6A inhibits KIF1C microtubule co-sedimentation. (**A**) Binding of full-length KIF1C to microtubules in the presence of Rab6A. In vitro synthesized $^{35}$S-myc-KIF1C was desalted, incubated with GTPγS-preloaded His-tagged Rabs (4.2 µM), and then with 0.8 µg/µl Paclitaxel stabilized pre-polymerized microtubules in 2.5 mM ADP and 0.5 mM GTPγS. Reactions were centrifuged through a 10% sucrose cushion and pellets were analyzed by scintillation counting. The fraction of full-length KIF1C cosedimenting with microtubules in the presence of the indicated Rabs is shown (error bars = SE [n ≥ 2]). (**B**) Rab6 affects KIF1C motor domain microtubule co-sedimentation in a concentration-dependent manner. KIF1C-355-His (160 nM) was incubated with increasing concentrations of His-Rab6A Q72L, and then with 2.1 µM Paclitaxel stabilized pre-polymerized microtubules in 2.6 mM ADP and 0.35 mM GTPγS. Reactions were centrifuged through a 35% sucrose cushion. Pellets were analyzed by fluorescent antibody immunoblot. (**C**) Rab6A affects the strong microtubule binding state of KIF1C. Purified KIF1C-355-His (80 nM) was incubated with His-Rab6A Q72L (4.86 µM) or BSA (7.6 µM), and then with increasing concentrations of microtubules in 2.6 mM AMP-PNP and 0.35 mM GTPγS. Samples were processed and analyzed as in **B**. (**D**) Rab6A affects the weak microtubule binding state of KIF1C. Purified KIF1C-355-His (160 nM) was incubated with His-Rab6A Q72L, His-Rab33, or BSA (3.42 µM), and then with increasing concentrations of microtubules in 2.6 mM ADP and 0.35 mM GTPγS. Data were fit using GraphPad Prism software. The fraction of motor sedimented is normalized to the amount of microtubules pelleted, determined by Coomassie blue staining.

The following figure supplement is available for figure 5:

**Figure supplement 1**. Rab6A inhibits KIF1C binding and motility.

## Rab6A affects the localization of KIF1C

To determine if the effects of Rab6A on KIF1C could also be detected in cells, we examined the localization of KIF1C constructs in HeLa cells using methanol fixation, which favors cytoskeletal structure (*Figure 6A*). While both full-length KIF1C and KIF1C-400 were localized on microtubules primarily at the cell periphery, KIF1C-350 localized along microtubules throughout the cell (*Figure 6A*). Unlike the KIF1C-350 construct, the KIF1C-400 construct contains the neck linker region that in KIF1A is important for regulating microtubule association and likely dimerization (*Hammond et al., 2009*; *Hirokawa et al., 2009a*). Thus, the localization observed is consistent with earlier findings. As KIF1C-350 displayed an easily monitored distribution and was localized to regions capable of maximal contact with Rab6A, we used this construct to examine the consequences of

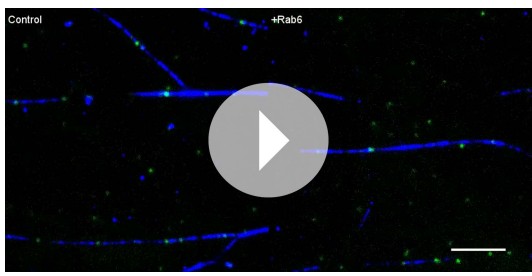

**Video 1.** Time-lapse imaging of KIF1C (green) motility on microtubules (blue) in vitro in the absence (right) or presence (left) of Rab6. More KIF1C motors can be seen under control conditions. More motors are moving. The time lapse covers 300 frames (12.3 s) and is sped up slightly (video is 10 s). Scale bar, 5 μm. (QuickTime; 11.6 MB).

concurrent Rab6A over-expression. KIF1C-350 wild-type or loop mutant proteins were expressed in Vero cells, with or without Rab6A. As shown in *Figure 6B*, both the KIF1C wild-type and loop mutant constructs localized to microtubules in the absence of Rab6A (top two rows). However, upon Rab6A co-expression, the overall distribution of wild-type KIF1C-350 over microtubules was lost and the staining was also less continuous (row 3, *Figure 6B,C*). Moreover, Rab6A only influenced the localization of the wild-type KIF1C and not the loop mutant (row 4, *Figure 6B,C*). These results were quantified using CellPro-filer (*Carpenter et al., 2006*) by determining the Pearson's correlation coefficient as a measure of the colocalization between KIF1C and microtubules (*Figure 6C*, top). Co-expression of Rab6A significantly reduced the correlation

of wild-type KIF1C-350 and microtubules by 33% (p < 0.001) but did not significantly affect the correlation between loop mutant KIF1C and microtubules. Importantly, the overall expression of KIF1C in co-transfected cells did not change in response to Rab6A, as the total intensity of KIF1C, quantified in paraformaldehyde fixed cells to capture the total KIF1C pool, was similar under all transfection conditions (*Figure 6C*, bottom). These experiments show that Rab6A influences microtubule association of the KIF1C motor domain in cells.

## KIF1C participates in intra- and post-Golgi vesicle transport

Rab6A is needed for overall Golgi structure and for transport of proteins to and from the trans-Golgi network (*Storrie et al., 2012*); KIF1C seems to perform similar roles (*Simpson et al., 2012*). We tested specifically whether KIF1C participates in protein transport from the Golgi to the cell surface using KIF1C siRNA which led to a greater than 90% depletion 72 hr post-transfection (*Figure 7—figure supplement 1A*). Previous work showed that KIF5B participates in this process and suggested that an additional kinesin contributes as well (*Grigoriev et al., 2007*). The plasma membrane delivery of Vesicular stomatitis virus (VSV) G glycoprotein in HeLa cells was monitored using a cell surface antibody-binding assay. Cells expressing YFP-VSV-G ts045 protein were held at 39°C to accumulate this temperature-sensitive protein in the ER, post-synthesis. Cells were then incubated at 32°C to release the block and permit cell surface delivery. VSV-G protein began to appear at the cell surface within 35 min after release of the block in control cells (*Figure 7—figure supplement 1B*). In cells depleted of KIF1C, however, cell surface delivery was significantly slowed (*Figure 7—figure supplement 1B*) and resulted in a 4.5-fold decrease in the overall delivery rate. Thus, KIF1C participates in cell surface delivery of proteins from the Golgi complex, consistent with work from other labs.

## KIF1C aids Rab6A-vesicle transport

As KIF1C participates in VSV-G transport, we explored the role of KIF1C in Golgi-derived vesicle transport using time-lapse epi-fluorescence microscopy of live cells expressing GFP-Rab6A, in the presence or absence of KIF1C (*Video 2*). GFP-Rab6A-labeled vesicles were segmented and tracked using u-track 2.1.0 (*Jaqaman et al., 2008*). Motion analysis was then used to categorize vesicles as being either confined (undergoing only localized random motion) or linearly moving.

While the absolute fraction of vesicles characterized as being 'confined' did not change in the presence or absence of KIF1C siRNA (not shown), confined vesicles in KIF1C-depleted cells occupied, on average, a twofold larger area when compared with the same population in control-depleted cells (*Figure 7A,C*). KIF1C depletion also affected Rab6A vesicles that were moving linearly: life histories of linearly moving vesicles showed that KIF1C depletion yielded more backwards movement events (*Figure 7B,E*) although no statistically significant differences in vesicle stalling were detected (0.084 vs 0.10 pauses per second, ± KIF1C siRNA, respectively). Similar findings were reported by *Grigoriev et al. (2007)* for neuropeptide Y vesicle motility in Rab6A-depleted cells.

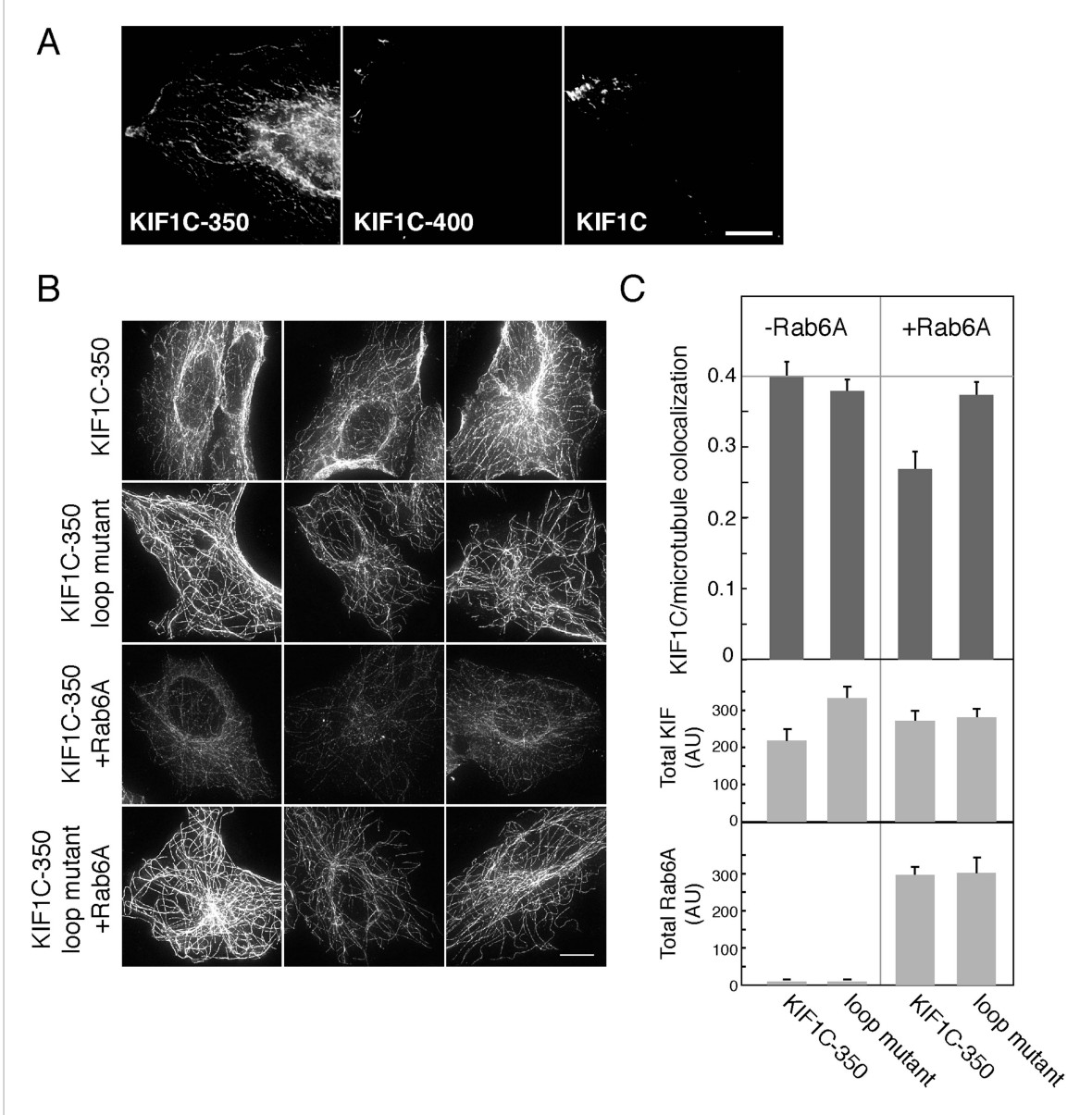

**Figure 6**. Rab6A inhibits KIF1C microtubule colocalization in cells. (**A**) Localization of KIF1C in MeOH-fixed HeLa cells transiently transfected with CFP-KIF1C-350, CFP-KIF1C-400, or full-length CFP-KIF1C. Scale bar, 10 μm. (**B**) Localization of CFP-KIF1C-350 or CFP-KIF1C-350 Loop 6/10 mutant ± Rab6A in MeOH-fixed Vero cells. Three examples are shown for each condition. Scale bar, 10 μm. Image levels were adjusted identically. (**C**) Mean KIF1C-microtubule co-localization measured by Pearson's correlation over KIF1C segmented objects from cells such as those shown in **B** (error bars = SE, >80 cells/condition). CFP-KIF1C-350 + Rab6A was statistically different from the other populations (p < 0.001). Below, average total fluorescence intensity of CFP-KIF1C and mCherry-Rab6A from similarly treated cells fixed in paraformaldehyde and quantified (error bars = SE, >100 cells/condition). Differences in total KIF intensity were not statistically significant between any of the populations (p > 0.05). Total Rab6A intensity was not statistically significant between the non-treated populations or between the treated populations (p > 0.05).

While the total distance traversed did not change significantly upon KIF1C depletion (data not shown), the mean speed of Rab6A vesicles increased by 31% (*Figure 7D*). However, this speed increase was not correlated with productive movement, as the number of directional changes along the principle axis of motion increased 74% upon KIF1C depletion (*Figure 7E*). Speed increases, loss of confinement, and loss of directionality upon KIF1C depletion indicate that the role of KIF1C is most evident when Rab6A vesicles are moving least—KIF1C promotes vesicle directionality. When vesicles are confined, KIF1C helps to maintain that confinement, and when vesicles are stalled, KIF1C prevents

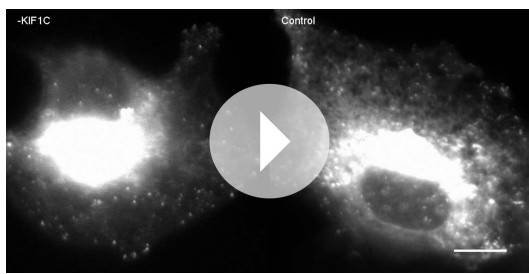

**Video 2.** Live cell imaging of pEGFP-Rab6 transfected Vero cells after KIF1C- (left) or control- (right) siRNA treatment. The large bright mass near the center is the Golgi and vesicles are seen as smaller punctate structures. The time lapse covers 71 frames (142 s) and is sped up 14 fold. Scale bar, 10 μm. (QuickTime; 5.7 MB).

backwards motion. An increase in vesicle directionality may be due to precluding the vesicle from binding a faster, less processive motor, or by influencing the tension on the vesicle, thereby changing the activity of other motors on the vesicle, and may come at the expense of overall vesicle speed.

It was not possible to achieve plasmid rescue of KIF1C levels after siRNA depletion as KIF1C over-expression and artificially induced recruitment also influenced Rab6A vesicle motility (data not shown, *Schlager et al., 2014*). Taken together, these results show that KIF1C is involved in both holding Rab6A vesicles within confined areas and also in maintaining the directionality of linearly moving Rab6A vesicles.

## Rab6A binding sites on KIF1C are required for the maintenance of normal Golgi structure

Depletion of KIF1C was previously reported to cause Golgi fragmentation in HeLa cells (*Simpson et al., 2012*). We reasoned that loss of Golgi ribbon structure may be related to KIF1C's ability to bind Rabs at both ends, which could in theory, be used to bind and link two adjacent membrane compartments. We therefore tested whether KIF1C's role in Golgi morphology maintenance correlated with its ability to bind Rab6A within the motor domain.

HeLa cells were transfected with either a control or KIF1C-targeting siRNA for 72 hr prior to fixation. After fixation, Golgi complexes were visualized using anti-p115 antibody. As expected, cells depleted of KIF1C displayed fragmented Golgi complexes that very likely represent peri-nuclear mini-stacks (*Figure 8A*, *Figure 8—figure supplement 1*). Upon rescue by siRNA-resistant, wild-type KIF1C plasmid transfection (24 hr prior to fixation), Golgi morphology returned to normal. In contrast, cells rescued with the full-length KIF1C motor domain loop mutant showed Golgi fragmentation similar to KIF1C depletion alone (*Figure 8A*). We also tested a construct, KIF1C K103A, that should not bind ATP (*Dorner et al., 1998*; *Li et al., 1998*) and unlike wild-type KIF1C, does not localize to the cell periphery (*Figure 8—figure supplement 2*) or to microtubules (*Figure 4—figure supplement 2*) but is still capable of binding to Rab6A (*Figure 4C*). Remarkably, KIF1C K103A rescued Golgi ribbon morphology in a manner similar to wild-type KIF1C rescue (*Figure 8*). Over-expression of these rescue constructs in cells treated with control siRNA did not significantly alter Golgi morphology (data not shown).

The effects of KIF1C depletion and subsequent rescue were quantified using CellProfiler (*Carpenter et al., 2006*) and custom Matlab algorithms (https://github.com/lee-ohlson-pfeffer/kif_golgi_fragmentation). To quantify the degree of Golgi fragmentation, we measured the percent of each cell's Golgi found in large objects (>4.11 μm); a high value would be seen for control cells. As expected, cells treated with KIF1C siRNA alone or rescued with the KIF1C 6/10 loop mutant showed a decrease in large Golgi objects. In contrast, the KIF1C wild-type and K103A constructs efficiently rescued Golgi morphology upon KIF1C depletion (*Figure 8*) despite a predominantly cytosolic localization for the K103A construct (*Figure 4—figure supplement 2*, *Figure 8—figure supplement 2*). Importantly, the motor domain loop mutation did not alter KIF1C localization, as both wild type and loop-mutant proteins localized largely at the cell periphery, indicating that the loop-mutant was well folded and able to migrate in an anterograde direction along microtubules (*Figure 8—figure supplement 2*). Over-expression of both wild-type KIF1C and KIF1C K103A increased the percent of large Golgi objects compared with control siRNA, which suggests that additional KIF1C may increase Golgi compactness. These results show that the sequences comprising the Rab6A binding site within the KIF1C motor domain contribute, in some way, to the maintenance of normal Golgi morphology and that KIF1C influences Golgi morphology, independent of its function as a motor.

An alternative means to assay Golgi ribbon stability is to test whether a protein influences the organization of Golgi mini-stacks in nocodazole-treated cells. Upon nocodazole addition, microtubules

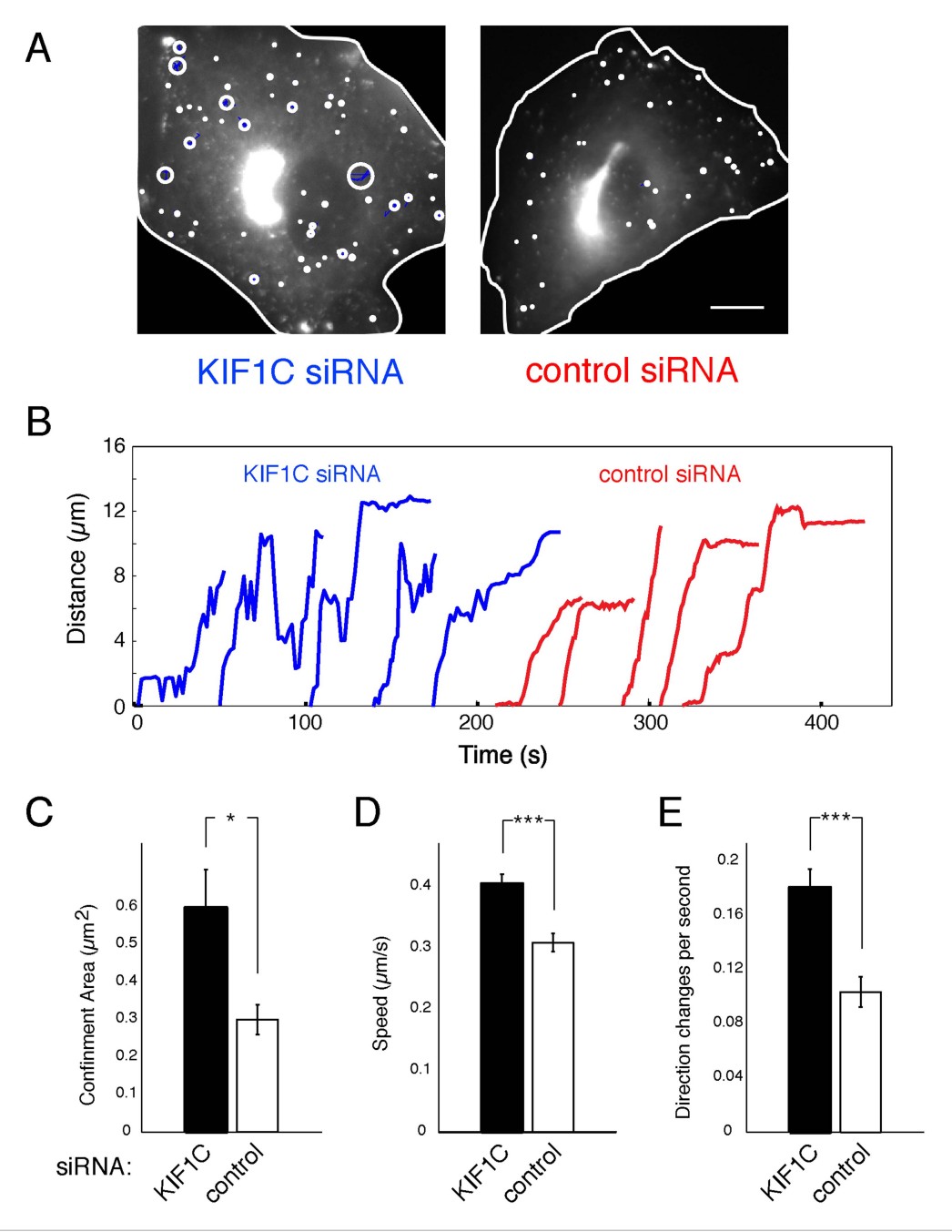

**Figure 7**. Loss of KIF1C affects Rab6A vesicle motility. Vero cells transfected with control- (30 cells) or KIF1C- (29 cells) siRNA followed by pEGFP-Rab6A (~300 vesicles imaged per cell). U-track 2.1.0 was used to segment, track, and characterize vesicles. Motion analysis was used to segment tracks into confined (>750 per condition) and linear populations (>70 per condition). (**A**) Traces of confined tracks in representative KIF1C- and control-siRNA cells. One frame from pEGFP-Rab6A videos (cells outlined in white) was overlaid with the identified confined tracks (blue). The confinement area for the random motion of each track is outlined (white circles). Scale, 10 μm. (**B**) Life history plots of linearly moving tracks from cells transfected with the indicated siRNAs. Five tracks were taken randomly for each population. (**C**) Quantitation of the confinement area as shown in **A**, error bars = SE (p < 0.05). (**D**) Quantitation of the speed of linear tracks as shown in **B**. The speed of each track was computed as the mean of the track's displacement divided by time between frames; error bars = SE (p < 0.0001). (**E**) Mean number of direction changes per second in linear tracks. Principle component analysis was used to find the main components of each track.
*Figure 7. continued on next page*

*Figure 7. Continued*

Direction changes were tabulated each time a track shifted from positive to negative along the main component excluding periods of pausing (speeds less than 0.1 µ/s), error bars = SE (p < 0.0001).
The following figure supplement is available for figure 7:

**Figure supplement 1**. Depletion of KIF1C impairs Golgi-to-cell surface transport.

begin to depolymerize and the Golgi is detected as characteristic mini-stacks that distribute throughout the cytoplasm (*Thyberg and Moskalewski, 1985*).

As shown in *Figure 9* (and *Figure 9—figure supplement 1*), Golgi ribbons in cells over-expressing wild-type KIF1C were somewhat protected from nocodazole-induced fragmentation. In contrast, expression of the KIF1C loop 6/10 mutant that binds microtubules (*Figure 4—figure supplement 2*) but shows diminished Rab6A binding (*Figure 4C*) did not protect the Golgi from nocodazole-triggered ribbon breakdown. Similarly, KIF1C lacking the C-terminal Rab binding domain also failed to protect the Golgi from nocodazole-induced fragmentation. In addition, the KIF1C K103A protein that can bind Rab6A at both ends but not microtubules, was able to protect the Golgi ribbon from disassembly (*Figure 9A* and *9B*). KIF1C E170A, mutated in a conserved kinesin microtubule binding site (*Woehlke et al., 1997*; *Grant et al., 2011*), was also able to protect the Golgi ribbon. This mutant showed diminished localization at the cell periphery compared with wild-type KIF1C (*Figure 8—figure supplement 2*).

To quantify these observations, we measured the percent of each Golgi found in large objects (>2.74 µm). These objects were smaller than those scored in siRNA rescue because the over-expression of wild-type KIF1C was not able to fully rescue nocodazole-induced fragmentation, despite a readily apparent rescue phenotype. Nocodazole treatment of control cells, as well as cells expressing KIF1C loop 6/10 or ΔCBD mutants, resulted in a decrease in the mean percentage of large Golgi objects. In contrast, wild-type KIF1C, K103A, or E170A mutant proteins protected the Golgi from fragmentation, even after nocodazole-triggered, microtubule depolymerization, as seen by the increase in percentage of Golgi detected in large objects. These data demonstrate that KIF1C has the capacity to affect Golgi morphology, independent of its motor stepping and microtubule binding activities.

Influencing Golgi morphology via sequences capable of Rab6 binding suggested that a small pool of KIF1C might form contacts with the Golgi. As noted earlier, KIF1C was first identified as a Golgi-localized motor protein (*Dorner et al., 1998*) but in addition to its abundant cytosolic pool, it has since been found at the cell periphery at podosome ends (*Bhuwania et al., 2014*; *Efimova et al., 2014*; *Kopp et al., 2006*; *Figure 6A* and *Figure 8—figure supplement 2*) and peri-centrosomally (*Schlager et al., 2010*; *Theisen et al., 2012*; *Figure 8—figure supplement 2*). Organelle association of predominantly cytosolic proteins can be difficult to detect. This challenge can be ameliorated by permeabilizing cells under conditions in which the cytosol is released, prior to fixation. For this purpose, we immersed coverslips in liquid nitrogen, which breaks plasma membranes and can reveal smaller pools of membrane-associated, cytosolic proteins (*Seaman, 2004*).

Upon expression of CFP-KIF1C in HeLa cells in the absence or presence of limited mCherry-Rab6A co-expression, deconvolution light microscopy in conjunction with liquid nitrogen permeabilization revealed that wild-type KIF1C, and more clearly, the non-motile KIF1C K103A mutant, form at least some contacts with the Golgi, monitored with antibodies to the trans Golgi GCC185 protein (left column) or exogenously expressed mCherry-Rab6A (right columns, *Figure 9C*). In the absence of nocodazole, residual KIF1C was distributed throughout the cell, with a clear perinuclear pool forming contacts with the Golgi, best detected at high magnification (middle rows). Golgi contacts were also detected at intermediate times of nocodazole treatment (45 min). KIF1C K103A also showed Golgi contacts (*Figure 9C*, bottom two rows). These experiments demonstrate that a small pool of KIF1C is appropriately situated to contribute to the process of stabilizing Golgi structure. Whether the mechanism of stabilization is direct or indirect, KIF1C is clearly important for normal Golgi structure maintenance.

## Discussion

The classical model for motor–cargo interaction consists of N-terminal domains interacting with the cytoskeleton and C-terminal domains interacting with cargo. A number of motors link to membranes via Rab GTPases, but few bind Rabs directly and none, to our knowledge, bind Rabs

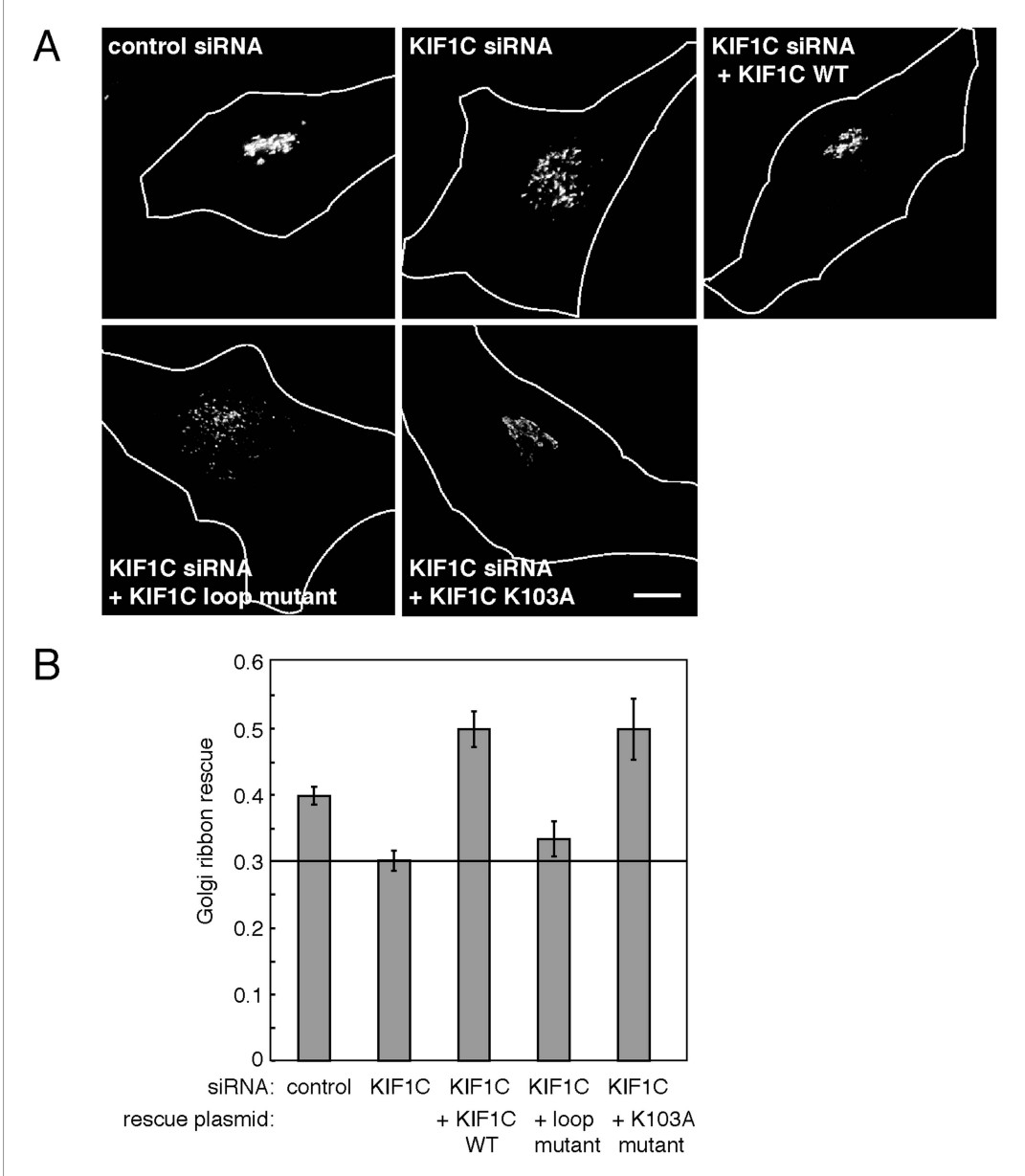

**Figure 8**. KIF1C's N-terminal Rab6A binding site is required for the maintenance of Golgi morphology. (**A**) Golgi morphology (p115) of HeLa cells transfected with control siRNA or KIF1C siRNA and the indicated rescue plasmid. Scale bar, 10 μm. (**B**) Golgi ribbon rescue, defined as the mean fraction of Golgi staining present as large objects (>4.11 μm), normalized by KIF1C intensity, quantified from cells such as those shown in **A** (bars = SE, >90 cells/condition). KIF1C wild-type rescue cells were statistically different from KIF1C depleted and loop mutant rescue cells but not those rescued with KIF1C K103A (p < 0.001).

The following figure supplements are available for figure 8:

**Figure supplement 1**. siRNA rescue shows that KIF1C's N-terminal Rab6A binding site is required for the maintenance of Golgi morphology.

**Figure supplement 2**. Full-length KIF1C mutant protein localization.

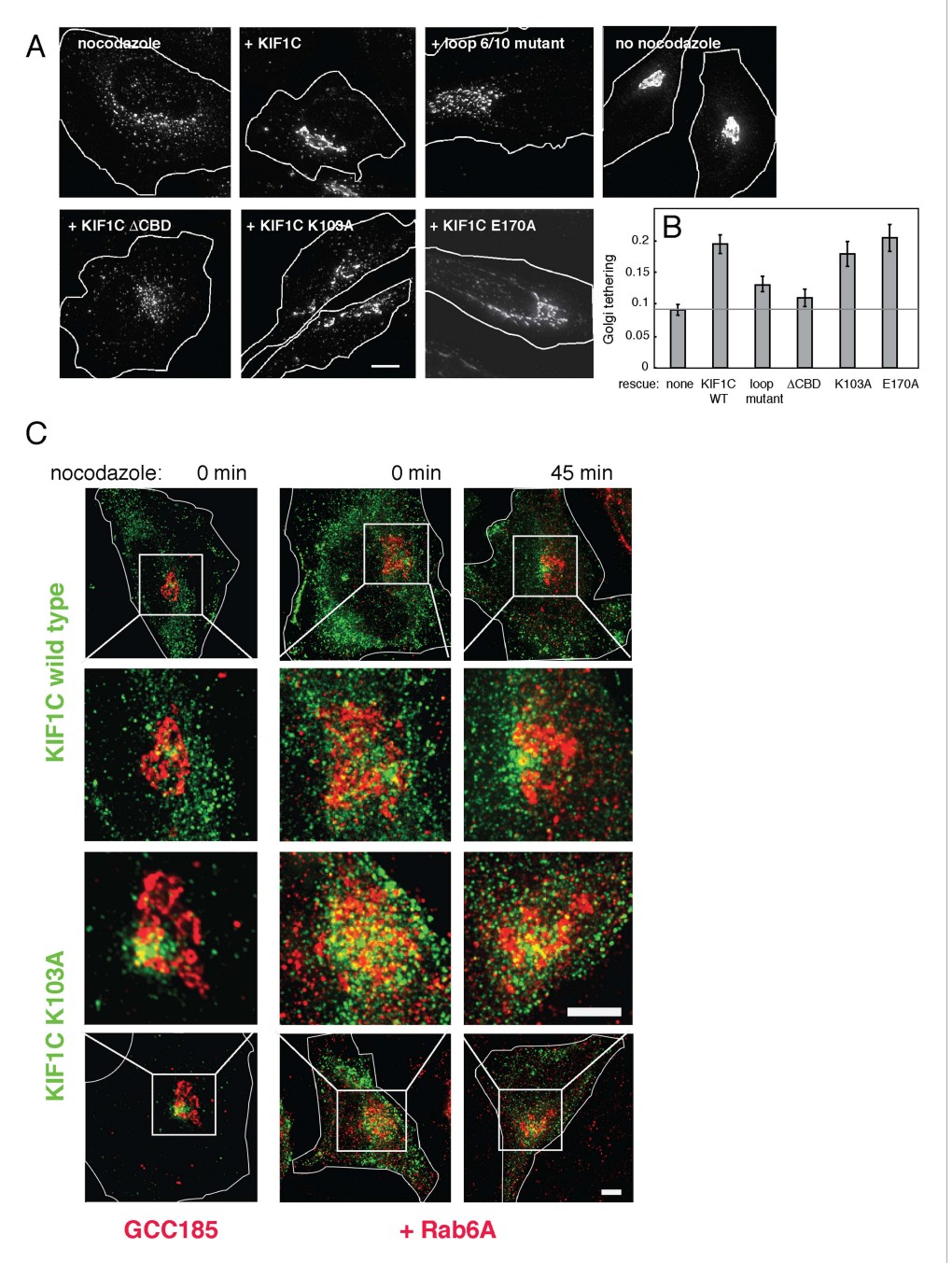

**Figure 9**. KIF1C over-expression stabilizes the Golgi in nocodazole treated cells. (**A**) Golgi morphology of HeLa cells incubated with or without nocodazole in the presence of the indicated over-expressed KIF1C construct. Scale bar, 10 μm. (**B**) Golgi structure maintenance, defined as the mean fraction of Golgi staining present in large objects (>2.74 μm), normalized by KIF1C intensity, in nocodazole-treated cells (bars = SE, >85 cells/condition). KIF1C wild type over-expressing cells were statistically different from KIF1C loop mutant and ΔCBD cells but not KIF1C K103A expressing cells (p < 0.01). (**C**) Perinuclear KIF1C forms contacts with the Golgi. CFP-KIF1C-transfected cells in the presence (left column) or absence (right columns) of co-transfected mCherry-Rab6A were incubated with nocodazole for the indicated times and treated with liquid nitrogen prior to fixation to release cytosolic proteins. KIF1C localization (wild type or K103A as indicated, green) was monitored in relation to the trans Golgi marker, GCC185 or to

*Figure 9. continued on next page*

*Figure 9. Continued*

the co-expressed mCherry-Rab6A (red). Magnified areas are indicated with a box and shown in the middle two rows. Scale bars, 5 μm.

The following figure supplement is available for figure 9:

**Figure supplement 1**. KIF1C over-expression stabilizes the Golgi in nocodazole-treated cells.

via their motor domain. We have shown here that Rab6A can bind directly to the KIF1C C-terminus ($K_D$ = 0.9 μM), consistent with the role of Rab GTPases as cargo adaptors. Surprisingly, we have also discovered a new mode of motor protein regulation: direct binding of a normally membrane-associated Rab GTPase to the KIF1C motor domain ($K_D$ = 0.2 μM). Our results show that Rab6A binds more tightly to the motor domain than to the cargo binding domain in vitro. However, in cells, the situation may be more complex as other proteins and/or KIF1C post-translational modifications may contribute to Rab6A's domain binding preferences.

Rab GTPase binding to the motor domain impairs KIF1C's ability to bind to microtubules, both in equilibrium binding assays and in cells. Indeed, Rab6A led to a greater than 10 fold decrease in the affinity of the motor for microtubules in both AMP-PNP and ADP states. Thus, KIF1C may have a choice: the motor domain can bind either microtubules or Rab6A on membranes with very similar affinities ($K_D$ 0.46 μM vs ∼ 0. 2 μM, respectively). This feature, combined with KIF1C's ability to interact with Rab GTPases via its C-terminus ($K_D$ = 0.9 μM), may enable KIF1C to transition from a transport vesicle motor to a protein capable of binding a Rab GTPase at both ends.

Rab6A may influence control on KIF1C-driven vesicle traffic in novel ways. If KIF1C is an auto-inhibited motor like conventional KIF5C (*Dietrich et al., 2008*) and KIF1A (*Hammond et al., 2009*), direct binding of Rab6A to the C-terminus may couple cargo binding with relief of motor protein auto-inhibition. Such a mechanism ensures that motor function is coupled to successful cargo engagement. Once on vesicles, additional Rab6A binding to the motor domain binding may regulate the load that other motors on the vesicle experience, by blocking KIF1C engagement with microtubules. This regulation may influence the speed, run length, and directionality of Rab6A vesicles, and may also play a role in KIF1C's confinement of non-moving vesicles.

In addition to the direct interaction between Rab6A and KIF1C described here, additional Rab6A effectors have been reported to interact at least indirectly with KIF1C. These include BICDR-1, which interacts in lysates and by yeast two-hybrid at the C-terminus of KIF1C (*Schlager et al., 2010*), and myosin IIA, which in lysates, interacts in the region before the C-terminus (residues 713–811; *Kopp et al., 2006*). Such interactions between Rab6, its effectors, and the C-terminus of KIF1C highlight the importance of the linkage between KIF1C and Rab6, all of which may strengthen the overall interaction of this motor with its cargo and provide an additional layer to regulate Rab6–KIF1C interactions. During neuronal development, BCDR-1 decreases significantly (*Schlager et al., 2010*). The ability of KIF1C to interact with Rab6A GTPase directly (as reported here) or via Rab binding proteins such as BCDR-1 suggests that KIF1C may have both constitutive and development-specific functions.

It would be difficult for a single Rab6A molecule to bind to both the KIF1C motor domain and to KIF1C's C-terminus, as both binding sites involve interaction with the Rab6A 'switch domains' that report the identity of the bound guanine nucleotide on the Rab protein surface. Multiple binding sites offer the possibility for Rab6A concentration-dependent effects on KIF1C, which will be of interest to explore in future experiments. In principle, KIF1C should be able to sense the concentration of Rab6A on transport vesicles, on the donor compartments from which they derive, or on the acceptor membrane once at their destination and choose between engaging microtubules or additional Rabs. Other adapters, such as BICDR-1, may also play a role in influencing how concentration affects each Rab6A binding site.

We have shown that sequences comprising Rab6A binding sites at both ends of KIF1C enable KIF1C to contribute to Golgi morphology maintenance, in a manner that is separable from its function as a motor. This explains the Golgi fragmentation phenotype seen here and previously upon KIF1C depletion (*Simpson et al., 2012*). Fibroblasts from mice lacking KIF1C did not appear to show a morphological Golgi defect (*Nakajima et al., 2002*), but such cells likely compensate by expressing

other proteins at higher levels and this subtle phenotype may have been missed. Myosin V is a well-characterized example of a motor protein that serves to tether bound cargo near the terminal actin web (*Wu et al., 2000*). In membrane traffic, tethers hold transport vesicles near their targets to permit engagement of SNARE proteins that mediate vesicle fusion. As KIF1C appears to be able to bind to Rabs directly at both its N- and C-termini, KIF1C might be employed similarly to help hold vesicles near the Golgi during the transition between microtubule based movement and docking at a membrane. The capacity of KIF1C to bind to Rabs at both ends also suggests a possible mechanism for KIF1C's role in maintaining Golgi morphology. Additional work will be needed to confirm these models for KIF1C function.

While motor function was not necessary for KIF1C's role in contributing to Golgi structure maintenance, cellular depletion of KIF1C had several additional, interesting consequences. KIF1C was needed both to confine relatively stationary vesicles and to sustain the momentum of moving vesicles. Rab6A vesicles lacking KIF1C move faster from frame to frame; they also change directions more often than vesicles in control cells. These results suggest that KIF1C has at least two functions: (1) to hold vesicles; (2) to increase transport directionality, which surely requires motor activity. The most visible role for KIF1C is to function as a motor, as seen by the motor's localization primarily at the cell periphery. Rab6A regulation of KIF1C could allow KIF1C to release from microtubules when bound to Rab6A at both ends. On vesicles, this might help to avoid vesicle traffic jams by allowing for backwards motion or adjust vesicle speed by allowing other motors to dominate vesicle motion. By aiding in directional motion and decreasing speed, it is also possible that KIF1C helps vesicles recognize their targets, as slower moving vesicles would have more time for the protein:protein interactions that drive vesicle docking and SNARE-mediated fusion.

Finally, it is important to remember that multiple motor proteins work in concert to transport vesicles. Rab6A participates in the motility of post-Golgi, exocytic transport vesicles (*Grigoriev et al., 2007*) in cooperation with the kinesin motor, KIF5B, and with cytoplasmic dynein via its effector, bicaudal D1 and D2 proteins (*Matanis et al., 2002*). Depletion of KIF5B did not fully block the motility of Rab6A vesicles in previous work (*Grigoriev et al., 2007*), thus, KIF1C likely functions in conjunction with these motors to drive Rab6A vesicle motility. The novel regulation by Rab6A of the KIF1C motor domain may allow Rab6A to shift the dynamics between these motors. The conventional KIF5B kinesin might play the primary role in moving Rab6A vesicles, while the interplay between KIF1C and dynein/dynactin might contribute to processivity.

In summary, this is the first example to our knowledge of a Rab GTPase (or any non-cytoskeletal protein) binding to a motor domain in a manner that has the potential to influence whether that protein chooses to bind to a membrane or a microtubule track. KIF1C may attach to membranes via Rab6A binding at the C-terminus. Once bound to membranes, the motor domain can (1) bind a microtubule for conventional vesicle motility, or (2) bind to another Rab6A molecule on the vesicle to allow other motors to drive vesicle transport, or (3) bind to adjacent Rab6A-containing membranes like the Golgi complex where it can hypothetically, be held. Competition for Rab6A on the vesicle will be high as the absolute number of Rab GTPases on an individual transport vesicle is likely to be small (~10 per vesicle; *Takamori et al., 2006*) and many effectors, including two domains of KIF1C, will be vying to bind to Rabs. Moreover, as Rab6A is abundant on Golgi mini-stacks, it is in theory, possible that KIF1C could also serve to hold these mini-stacks together to maintain normal Golgi ribbon structure. This KIF1C-driven mechanism could explain how depletion of KIF1C leads to increased Golgi fragmentation that can be rescued by full-length KIF1C independent of motor activity but dependent on Rab6A binding domains. Finally, because the motor domain can potentially choose between Rab6A or microtubules, KIF1C could play a role during transport vesicle formation: when the tip of a newly forming transport vesicle projects away from a Rab6A-enriched membrane surface, the motor could instead latch onto microtubule filaments. KIF1C could then contribute to Rab6A vesicle motility en route to the cell surface. Future experiments will clarify how the Rab GTPase-assisted gymnastics accomplished by KIF1C contribute to membrane traffic within and beyond the Golgi complex.

## Materials and methods

### Cell culture and transfections

HeLa and Vero cells were cultured in Dulbecco's modified Eagle's medium and transfected with siRNA and plasmids as described (*Aivazian et al., 2006*). Rabbit anti-KIF1C antibody and goat

anti-rabbit-horseradish peroxidase were from Cytoskeleton, Inc. (Denver, CO) and Bio-Rad (Hercules, CA), respectively.

## Immunoprecipition

HEK293 cells were transfected with KIF1C and Rab6A constructs 48 hr prior to cell lysis in 50 mM Hepes, pH 7.4, 150 mM NaCl, 1% CHAPS, and protease inhibitors (cOmplete, EDTA free, Roche, Indianapolis, IN). Clarified cell lysate was incubated with llama anti-GFP binding protein (*Rothbauer et al., 2008*) conjugated to NHS-activated Sepharose 4 Fast Flow (GE Healthcare Biosciences, Pittsburgh, PA). After washing, the bound fraction was eluted in sample buffer and analyzed by immunoblot with mouse anti-Myc (9E10) or rabbit anti-Rab6A antibody (Santa Cruz Biotechnology, Dallas, TX).

## Rab binding

Purification of Rab GTPases was described (*Aivazian et al., 2006*). GST-tagged KIF1C constructs were expressed in bacteria and homogenized in 20 mM Tris, 400 mM NaCl, 1 mM DTT, pH 7.4. Clarified homogenates were bound to glutathione-Sepharose and eluted in 20 mM glutathione. Purified Rabs (10 µM) were preloaded with $^{35}$S-GTPγS, $^3$H-GDP, or cold nucleotide (for immunoblot analysis) in 50 mM HEPES pH 7.4, 150 mM KCl, 10 mM EDTA, 0.1% BSA, 1 mM DTT, and 100 µM nucleotide for 10 min at 37°C; 20 mM MgCl$_2$ was then added. Glutathione–Sepharose beads were incubated with 1 µM Rab and 15 µM GST or GST-KIF1C for 1 hr at room temperature in 20 mM HEPES pH 7.4, 150 mM KCl, 4 mM MgCl$_2$, 0.1% BSA, and 0.2% Triton X-100 (Buffer A). Beads were washed 3× and scintillation counting (LS 6500, Beckman Coulter, Inc., Indianapolis, IN) determined Rab bound. Alternatively, beads were eluted with 20 mM glutathione, pH 7.5 and analyzed by fluorescent immnuoblot using His-tag Rabbit antibody (Cell Signaling, Danvers, MA) and Alexa Flour 647 Goat anti-Rabbit antibody (Life Technologies, Grand Island, NY). Fluorescence signal was captured using a Typhoon imager (GE Healthcare Biosciences) and analyzed by ImageJ (*Schneider et al., 2012*).

## KIF1C in vitro translation and Rab binding

Myc-KIF1C constructs synthesized using TNT Quick Coupled Transcription/Translation System (Promega, Madison, WI), optimally 11.25 nM, were incubated with GTPγS- or GDP-loaded, GST-Rabs (0.2–5 µM) in Buffer A plus 1 mM DTT, and 100 µM GTPγS or GDP for 1 hr, 25°C, added to glutathione-Sepharose at 25°C, 1 hr and washed 3× (Buffer A +1 mM DTT, 400 mM NaCl) before elution with 20 mM glutathione, pH 7.5. Eluates were immunoblotted with mouse anti-myc antibody (9E10).

## KIF1C motor domain expression in *Escherichia coli*

KIF1C motor domain (residues 1–349) was fused to murine KIF5C residues 329–334 followed by 6×His tag (*Nitta et al., 2004*) and purified based on *Romberg et al. (1998)*. After expression in Rosetta 2 cells using 0.5 mM isopropyl B-*p*-thiogalactopyranside for 16 hr at 16°C, cells were suspended in Buffer B (50 mM NaPO$_4$, pH 7.4, 15 mM imidazole, 250 mM NaCl, 1 mM MgCl$_2$, 25 µM ATP, protease inhibitors) and disrupted by Emulsiflex C-5 (Avestin, Ottawa, ON). Clarified lysate was incubated with Ni-NTA (Qiagen, Santa Clarita, CA) for 1.5 hr at 4°C and after washing with Buffer B was eluted with Buffer B + 200 mM imidazole. The eluate was diluted fivefold with Buffer C (30 mM Hepes, pH 7.4, 1 mM MgCl$_2$, 1 mM EGTA, 25 µM ATP) and applied to HiTrap Q FF (GE Healthcare), washed with Buffer C + 100 mM NaCl, before being eluted by gradient to 500 mM NaCl in Buffer C. Binding of the purified components was assayed in the same manner as in vitro translation-synthesized constructs.

## Radioactive KIF1C Microtubule binding

Full-length myc-KIF1C synthesized in vitro with $^{35}$S-methionine (EasyTag, Perkin Elmer, San Jose, CA) was desalted and incubated with GTPγS-preloaded Rabs (4.2 µM) for 1 hr in Buffer A (−BSA), 1 mM DTT, 2.5 mM ADP, 0.5 mM GTPγS. Microtubules (0.8 mg/ml), polymerized in 80 mM PIPES, 1 mM MgCl$_2$, 1 mM EGTA, pH 6.8 (BRB80), 1 mM DTT, 1 mM GTP, 10% DMSO, spun through a 40% glycerol cushion, and resuspended in BRB80, 1 mM DTT, 0.2 mM Paclitaxel (Cytoskeleton, Inc.), were incubated with the KIF1C-Rab complexes for 1 hr before being spun through a 10% sucrose, 20 µM Paclitaxel, 1 mM DTT, at 65K for 5 min (Optima TLX, Beckman Coulter, Inc., Indianapolis, IN). Scintillation counting and SDS-PAGE and radiography using a Typhoon imager (GE Healthcare Biosciences) revealed the amount of $^{35}$S-labeled-KIF1C in fractions.

## Motor domain microtubule binding

Purified His-tagged motor domain was incubated with Rabs in BRB80, 1 mM DTT, 0.1 mg/ml BSA and indicated nucleotides for 30 min at room temperature with agitation. Microtubules were added for another 30 min. Reactions were centrifuged through a 35% sucrose cushion at 65K for 20 min (Beckman Coulter, Inc.). Pellets were visualized by SDS-PAGE and immunofluorescence using a Typhoon imager (GE Healthcare Biosciences) and analyzed by ImageJ (*Schneider et al., 2012*).

## KIF1C localization

CFP-KIF1C constructs were transfected into Vero or HeLa cells in the absence of presence of mCherry-Rab6A 24 hr before fixation. To measure total cell fluorescence intensity, cells were fixed with paraformaldehyde (3.7%, RT, 15 min) and permeabilized with 0.1% Triton X-100. To observe motor-microtubule localization, cells were permeabilized with MeOH (100%, −20°C, 4 min). To observe motor-Golgi localization, cells were treated with liquid nitrogen to permit release of cytosolic proteins, fixed with paraformaldehyde (3.7%, RT, 15 min) and permeabilized with 0.1% Triton X-100 (*Seaman, 2004*). Cells were stained with chicken anti-GFP (Abcam, Cambridge, MA), rabbit anit-Rab6A (GeneTex, Irvine, CA), mouse anti-GCC185 (produced for us by Cocalico Biologicals, Reamstown, PA), and mouse anti-tubulin (Sigma–Aldrich, St. Louis, MO) antibodies and visualized using an Olympus IX70 microscope with a 60× 1.4 NA Plan Apochromat oil immersion lens (Olympus, Center Valley, PA) and a charge-coupled device camera (CoolSNAP HQ, Photometrics, Tucson, AZ). Maximum intensity projections were generated using softWoRx 4.1.0 software (Applied Precision, Issaquah, WA). ImageJ (*Schneider et al., 2012*) was used to measure the total fluorescence intensity of traced cells. CellProfiler (*Carpenter et al., 2006*) was used to segment MeOH-treated cells and measure the Pearson's correlation coefficient between tubulin and KIF1C intensity over KIF1C-segmented objects. Two-sample t-test was used to determine significance.

## KIF1C single molecule assay

Full-length His-KIF1C was purified as described (*Hirokawa and Noda, 2001*). Coverslips coated with anti-β tubulin antibody and blocked with 0.75% Pluronic F-127 (Sigma–Aldrich) were incubated with 7.5 µg/ml rhodamine-labeled pre-formed microtubules + 15 µM Paclitaxel and blocked by casein. GTPγS-loaded Rab6A (15 µM) or exchange buffer was mixed with 10 nM KIF1C plus rabbit anti-KIF1C antibody (Cytoskeleton, Inc.) and Dylight 649 goat anti-rabbit Fab fragment (Jackson ImmunoResearch, West Grove, PA). Samples were visualized using a Nikon Ti-E inverted microscope with a 100× 1.49 NA APO-TIRF lens and an EMCCD (Andor iXON+; Andor technology, South Windsor, CT). Motors were segmented using Spot Detector (*Olivo-Marin, 2002*) and tracks were found using u-track (110523) (*Jaqaman et al., 2008*). The Wilcoxon rank sum test was used to determine statistical significance. Samples were visualized in series such that proteins in each condition were on the coverslip for similar amounts of time. Moving motors were defined as those moving >0.12 µm/s for a distance of >0.2 µm that were tracked for >0.9 s. In these experiments, a large pool of motors was immobile on microtubules. However, the median life span of this pool was 0.7 s, suggesting that most motors do not accumulate on microtubules but rather bind and release. Importantly, both immobile and motile pools of KIF1C were susceptible to Rab6A action. A small pool of motors (6%) was long-lived (± Rab6A); this pool did not increase over the time of these experiments (∼15 min). Auto-inhibition likely explains why only 2% of microtubule-bound motors showed motility in these experiments.

## Rab6A vesicle motility analysis

Vero cells were transfected with siRNA (72 hr total) followed by pEGFP-Rab6A (for the final 36 hr). Coverslips were transferred to a 37°C heated stage in Leibovitz's L-15 medium (Invitrogen, Grand Island, NY) and filmed using a Nikon Eclipse 80i microscope using a 100× numerical aperture 1.40 plan apochromat oil immersion objective and an EMCCD (Andor technology). Vesicles in cropped stacks were segmented and tracked using u-track 2.1.0 (*Jaqaman et al., 2008*). Track parameters were not normally distributed and the Wilcoxon rank sum test was used to determine statistical significance.

## VSV-G transport assay

HeLa cells were transfected with either a KIF1C (CCUCAUGGAC UGUGGAAAUUU) or a non-targeting siRNA (GUUCAAUAGGCUUACUAAUUU) (Thermo Scientific, Lafayette, CO) for 20 hr followed by pVSV-G-YFP (ts045) for 2 hr at 37°C. VSV-G was accumulated in the ER (39°C, 16 hr), incubated at 32°C, then washed with ice-cold PBS, blocked with 0.2% BSA (30 min) and incubated with mouse anti-VSV-G (8G5F11) (1 hr, 4°C). Cells were scraped into RIPA buffer with protease inhibitors (Roche); the post-nuclear supernatant was immunoblotted with goat anti-mouse horseradish peroxidase (BioRad, Hercules, CA); bound anti-VSV-G antibody was normalized to total VSV-G.

## Golgi morphology analysis

HeLa cells were transfected with siRNA (72 hr total) followed by CFP-tagged rescue constructs 24 hr before fixation or transfected with CFP-tagged rescue constructs (24 hr) and then incubated with nocodazole (5 μg/ml, 1 hr, 37°C). Rescue constructs were made insensitive to siRNA by replacing seven nucleotides while retaining coding identity (GACCTCATGGACTGTGGAAAT to GATTTAATG GATTGCGGTAAC). To observe Golgi morphology, cells were fixed with paraformaldehyde (3.7%, RT, 15 min) and permeabilized with 0.1% Triton X-100. Cells were stained with chicken anti-GFP (Abcam) and mouse anti-p115 (mouse ascites) antibodies and visualized using an Olympus IX70 microscope with a 40× 1.35 NA Apochromat oil immersion lens (Olympus) and a charge-coupled device camera (CoolSNAP HQ). Maximum intensity projections were generated using softWoRx 4.1.0 software (Applied Precision). Cells and Golgi were segmented using CellProfiler (*Carpenter et al., 2006*). Golgi complex morphology was scored for each cell as the percent of p115-positive structures defined as 'large' (4.11 μm$^2$ for siRNA and 2.74 μm$^2$ for nocodazole-treated cells) and normalized by the mean KIF1C intensity using Matlab (Mathworks, Natick, MA; https://github.com/lee-ohlson-pfeffer/kif_golgi_fragmentation). The two-sample t-test was used to determine statistical significance.

## Acknowledgements

This research was funded by an NIH grant (DK37332; www.nih.gov) and an American Diabetes Association grant (7-12-MN-67; www.diabetes.org) to SRP and fellowships from the National Science Foundation (www.nsf.gov) to PLL and NIH 5F32GM088980 to MBO.

## Additional information

### Competing interests

SRP: Reviewing editor, *eLife*. The other authors declare that no competing interests exist.

### Funding

| Funder | Grant reference | Author |
|---|---|---|
| National Institutes of Health (NIH) | DK37332 | Suzanne R Pfeffer |
| American Diabetes Association | 7-12-MN-67 | Suzanne R Pfeffer |
| National Institutes of Health (NIH) | 5F32GM088980 | Maikke B Ohlson |
| National Science Foundation (NSF) | Graduate Research Fellowship | Peter L Lee |

The funders had no role in study design, data collection and interpretation, or the decision to submit the work for publication.

### Author contributions

PLL, Conception and design, Acquisition of data, Analysis and interpretation of data, Drafting or revising the article; MBO, Acquisition of data, Analysis and interpretation of data; SRP, Conception and design, Analysis and interpretation of data, Drafting or revising the article

### Author ORCIDs

Peter L Lee, http://orcid.org/0000-0002-5553-9041
Suzanne R Pfeffer, http://orcid.org/0000-0002-6462-984X

# Additional files

## Major datasets

The following previously published datasets were used:

| Author(s) | Year | Dataset title | Dataset ID and/or URL | Database, license, and accessibility information |
|---|---|---|---|---|
| Kikkawa M, Hirokawa N | 2006 | KIF1A head-microtubule complex structure in adp-form | http://www.pdb.org/pdb/explore/explore.do?structureId=2HXH | Publicly available at RCSB Protein Data Bank (2HXH). |
| Nitta R, Okada Y, Hirokawa N | 2008 | Crystal Structure of the Kif1A Motor Domain Before Mg Release | http://www.pdb.org/pdb/explore/explore.do?structureId=2ZFI | Publicly available at RCSB Protein Data Bank (2ZFI). |
| Walden M, Jenkins HT, Edwards TA | 2011 | Crystal structure of D. Melanogaster Rab6 GTPase bound to GMPPNP | http://www.pdb.org/pdb/explore/explore.do?structureId=2Y8E | Publicly available at RCSB Protein Data Bank (2Y8E). |

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
