## [Decision Letter]

[Editors’ note: a previous version of this study was rejected after peer review, but the authors submitted for reconsideration. The first decision letter after peer review is shown below.]

Thank you for choosing to send your work entitled “KIF1C kinesin: A Rab GTPase-regulated vesicle motor and Golgi tethering protein” for consideration at *eLife*. Your full submission has been evaluated by Randy Schekman (Senior editor and Reviewing editor) and 3 peer reviewers, and the decision was reached after discussions between the reviewers. We regret to inform you that your work will not be considered further for publication.

Reviewer 2, Francis Barr, has agreed to reveal his identity. The other reviewers have chosen to remain anonymous.

The reviewers agree that the main finding that Rab6 interacts with both KIF1C motor and C-terminal domains and appears to regulate activity is interesting and important. However, the data in support of this conclusion are not strong. There are many clear controls that could have been done (e.g. the effect of Rab6 on the KIF-1C loop mutants interaction with MTs).

*Reviewer 1*:

In this work, Lee et al. identify an interaction between KIF1C and the small GTPase Rab6a and suggest that this interaction serves to regulate the activity of this motor. In particular, they focus on a new interaction that they describe between the KIF1C motor domain and Rab6, documenting the region of interaction and effects on MT binding and motility. They then do in vivo experiments to show effects of KIF1C depletion or certain overexpression experiments on Golgi distribution and vesicle transport. Overall, there is a great deal of work contained in this paper (11 experimental main figures and supplement). However, I have reservations regarding the publication in *eLife*. First of all, the in vitro effects of the Rab6 on KIF1C microtubule interaction are relatively small (∼50%), which makes one wonder about the in vivo relevance. The GDP/GTP selection of Rab6 is also weak (50%; Figure 3). Also, given that this is a core of the study, I thought that many of these in vitro experiments were not thoroughly documented as described below. I also thought that in vivo experiments did not convincingly establish a role for the Rab6 nucleotide switch on regulating KIF1C motor/microtubule binding function in vivo. Thus, while the data is interesting individually, I did not find that the paper as a whole conveyed a compelling story, or at least at the level required for an *eLife* publication.

A major defect in the paper was the characterization of in vitro effect of Rab6 on KIF1C.

Figures 6 and 7 (the core of the paper for showing effects on microtubules) should be done with the same protein, and ideally without a KIFC-KIF5 fusion. Ideally, the authors should try baculovirus expression to have a well-behaved protein that can be used in both sets of experiments:

a) The microtubule binding experiments should be done by performing a microtubule binding curve to measure Kd and should be done with control, Rab6-GDP and Rab6GTP.

b) The Figure 7 motility experiments are also suboptimal. First of all, they were done with an antibody and secondary antibody, which can result in crosslinking, and this is just very complicated to interpret. Direct labeling (e.g SNAP or Halo) would be better. Second, there is no measurement of velocity or processivity. Third, Rab6 should be tested for effects again in the GDP and GTP states. Fourth, the very low number of moving motors is troublesome; suggested to be due to autoinhbition but not really followed up. Fifth, it would be good to establish whether this KIF1C construct is monomeric or dimeric (or contains aggregates). Sixth, would be good to label Rab6 to see if it is co-bound to the motors on the microtubule (or not).

*Reviewer 2*:

In this study Pfeffer and coworkers describe and characterize the interaction of Rab6 with KIF1C, and describe a role in Golgi-associated transport. The demonstration that Rab6 can directly interact with and regulate the kinesin motor activity is an important finding that would justify publication in *eLife*. I have a few major comments.

1) The authors provide clear evidence for the interaction of Rab6a with the c-terminal region of KIF1C. This is highly specific for the GTP-bound form consistent with the proposal that KIF1C is a Rab6 effector. Surprisingly deletion of this region did not abolish Rab6 binding and the authors identify a second binding site in the n-terminal motor region. There is preferential binding to Rab6-GTP but not to the same extent as the CBD site. It would be interesting to know how the CBD affinity compares to the motor site ∼0.2µM. Does the Rab6 GTP specificity data indicate that CBD binds more tightly? The relationship between the two sites is important for understanding how Rab6 is recruited. If one site has a much higher affinity than the other then one would expect it to be occupied first, having important implications for the system.

2) By preventing motor activity with the K103A mutation, KIF1C is converted from a dynamic motor/tether to a static tether. When overexpressed in cells this construct may suppress Golgi ribbon defects, but this is not the same as proving motor activity is not normally required for KIF1C function in the system. For example in the speed and direction change assays in Figure 9 one would imagine the motor inactive KIF1C fails to rescue these aspects of Rab6 vesicle movement. The K103A rescue also looks slightly different to the wild-type KIF1C condition in the Golgi ribbon assay, so it clearly does not rescue completely. This possibility should be mentioned since it is relevant for the discussion of whether the motor domain interaction is a way of creating a Golgi ribbon tether or perhaps a form of motor activity regulation.

I wasn't entirely sure of the argument ion the subsection headed “Rab6A-KIF1C interaction is required for maintenance of normal Golgi structure” regarding the wild type and K103A mutant. If wild-type KIF1C fails to rescue, then surely the fact a mutant does rescue implies there is an important difference between the two relevant for normal function.

3) Does the ADP/ATP state of the motor domain make any difference to the Rab6A binding shown in Figure 5? This is relevant for understanding the relationship between kinesin-microtubule interaction and Rab6A binding and the data in Figures 6 and 7.

4) Do the KIF1C loop mutants have microtubule-stimulated ATPase activity? This should be shown or mentioned if it has been tested.

*Reviewer 3*:

In this study, Lee et al. build on their earlier finding that KIF1C binds to active Rab6, and explore the consequences of this interaction. This motor has previously been implicated in organization of the Golgi as its depletion results in Golgi fragmentation. They first show convincingly that the tail and head of KIF1C each have binding sites for Rab6-GTP. Using the knowledge that the head of KIF1A, for which a structure is available, does not bind Rab6, they make chimeras to identify the Rab6 binding region. They find that substitution of loops 6 and 10 from KIF1A into KIF1C abolishes Rab6 binding, but does not compromise microtubule binding of the head domain alone. Next, they report that Rab6 binding to the motor domain blocks both its ability to bind microtubules and in vitro motor motility. Next, they explore the role of KIF1C in vivo and find that siRNA knockdown results in observable differences in confinement, speed and directionality of pEGFP-Rab6A puncta. Next, the transit time of YFG-VSV-G-ts045 from the ER to the cell surface is found to be delayed in KIF1C siRNA cells. Additionally, the ribbon organization of the Golgi is disturbed in KIF1C siRNA cells, a phenotype that can be suppressed by KIF1C expression, and not by the loop 6/10 mutant, but, significantly, can be rescued by a KIF1C mutant that binds Rab6 but not microtubules. In support of this, nocodazole induced dispersion of the Golgi could be partly rescued by over-expression of KIF1C, and to a lesser degree by mutants that cannot bind Rab6.

This is an interesting and well conducted study that provides evidence that Rab6 binds the KIF1C motor head, and that this interaction is of physiological importance in microtubule binding, motor activity and Golgi tethering.

There seem to be a couple of issues:

1) A key result involves switching loops 6 and 10 out of KIF1C and replacing them with the corresponding ones from KIF1A. This negative result is interpreted as showing that the loops bind Rab6. There are other possible interpretations, most notably that the conformation is disturbed. Obviously, if the converse switch of loops 6 and 10 from KIF1C into KIF1A induces Rab6 binding, this would be excellent evidence. Failing this, the docked structure shown in Figure 5 should reveal critical residues in the loops that could be mutated to make a stronger case. The switched loop construct is used throughout the study—if loops 6 and 10 cannot be shown to be directly involved in Rab6 binding, this does not take away from the rest of the story, as this construct had a clear defect in Rab binding.

2) The data are consistent with an interaction between Rab6 and KIF1C based on coIP of over-expressed proteins, and in vitro studies. This is perhaps a bit surprising as the images in Figure 8 and Figure 9 suggest that over-expressed KIF1C is almost completely peripheral, whereas over-expressed Rab6 is almost all perinuclear. Can the endogenous proteins be seen to coIP or colocalize? Wouldn't a consequence of their model suggest that over-expression of Rab6 should drive KIF1C to the Golgi complex? This issue brings into question whether Rab6 is in fact the correct Rab protein that endogeneous KIF1C interacts with.

[Editors’ note: what now follows is the decision letter after the authors submitted for further consideration.]

Thank you for sending your work entitled “KIF1C kinesin: A Rab GTPase-regulated vesicle motor and Golgi tethering protein” for consideration at *eLife*. Your article has been favorably evaluated by Randy Schekman (Senior editor and Reviewing editor) and 3 reviewers.

I took the liberty of replacing one of the previous reviewers of your manuscript with a new expert and consulted two who had already commented on your last submission.

We discussed the comments of the three reviewers before we reached this decision, and I assembled the following comments to help you prepare a revised submission.

There continues to be a considerable degree of concern that you have not been able to colocalize KIf1C and Rab6A on Golgi membranes, and that without such a demonstration, the claim that KIF1C serves as a Golgi tether is in doubt. In the consultation session, there was discussion about further attempts to document this colocalization with other antibodies, but there was some feeling that this has not been successful in other laboratories and may not be feasible at this time. Under the circumstances, the view is that you should alter your title and conclusions to remove the tether assignment for KIF1C. Such a modification would be acceptable to the referees. However, you may wish to make one more attempt at colocalization, which if successful would considerably improve the impact of your work. Let us know how you wish to proceed.

*Reviewer 1*:

This paper describes the interaction between kinesin-3 KIF1C and the small GTPase Rab6. The authors show that Rab6A can specifically interact both with the motor and the tail domain of KIF1C. The authors suggest that the function of this interaction is two-fold: to mediate transport of Rab6A vesicles to the cell periphery, and to act as a Rab6A-specific inter-membrane Golgi tether.

Participation of KIF1C in the microtubule plus end-directed transport of Rab6A-decorated vesicles has already been documented (Schlager et al., EMBO J, 2010; Schlager et al., Cell Reports 8(5):1248-56, 2014; note that the second paper is recent and is not cited). The finding that a domain in KIF1C tail binds to Rab6A certainly helps to understand how the motor is recruited to the vesicles, but by itself does not seem to justify publication in *eLife*.

A more surprising claim in this study is that Rab6A can also bind to the KIF1C motor domain, prevent its interaction with microtubules and convert it into an inter-membrane tether. The biochemical data in the revised paper seem mostly reasonable. However, I have strong reservations about the cell biological data. If KIF1C serves as a Rab6A-dependent, functionally relevant Golgi tether—a protein that can connect vesicles to Golgi stacks or promote association between different Golgi stacks, like the authors suggest—this kinesin must be detectable at the Golgi. However, the paper does not contain a single image showing the Golgi localization of KIF1C, either endogenous or overexpressed. For example, in Figure 9, the authors show data indicating that overexpressed KIF1C or KIF1C 103A mutant suppress nocodazole-mediated Golgi dispersion. It would seem that KIF1C should be detectable at the Golgi in these conditions, especially as microtubules are not present and should not compete with Rab6A for binding. In the rebuttal letter, the authors indicate that there are no antibodies that can detect endogenous KIF1C. This statement does not seem accurate, because two papers (Theisen et al., Dev Cell 2012, Efimova et al., J Cell Sci 2014) successfully used KIF1C antibodies from Cytoskeleton to stain endogenous KIF1C (although no specific accumulation of KIF1C at the Golgi seemed to be present in the images in these papers). As an alternative to endogenous staining, a better effort could be made with tagged KIF1C and tag-specific antibodies. Ideally, the authors should show that there is a KIF1C pool at the Golgi, and that this pool is lost when Rab6A is depleted. If no KIF1C can be detected at the Golgi, then the most trivial explanation is that its levels at this membrane organelle are very low, bringing the “tether” model into question.

Other comments:

Figure 5—figure supplement 1: The data do not look convincing, the percentage of moving motors is extremely low, which in this type of assay typically indicates misfolding of the kinesin motor domain. Also the effect of more than a 1000-fold (!) excess of Rab6A on microtubule binding is weak.

Figure 6: Overexpressed Rab6A should be mostly present on the Golgi and vesicles. If the affinity between Rab6A and KIF1C is sufficiently high to “pull the motor off” the microtubules, the motor should relocalize to the Rab6A-positive membranes, an effect that should be detectable in PFA-fixed cells.

Figures 6 and 7: The effect of KIF1C depletion on the vesicle movement is mild, in agreement with published data, and is likely explained by participation of additional kinesins such as KIF5B. It is therefore unclear why the effect on VSV-G secretion is so strong. The authors say that rescues were not possible. Can they exclude that it is an off-target effect of the siRNA (for example, by showing knockdown with another siRNA)? Using at least two different siRNAs to show a phenotype seems now standard in the field.

*Reviewer 2*:

In this study Pfeffer and coworkers show that the kinesin KIF1C is regulated by the small GTPase Rab6. I felt the original submission was an important piece of work that could have been published in *eLife*. There were some issues that needed to be addressed and most of these have been dealt with in the revised submission.

I think it is important to clarify what the authors have found. First there is the identification and characterisation of two Rab6 binding sites in KIF1C. One of these sites lies within the kinesin motor domain. The authors provide evidence that this site is important for regulating KIF1C using a combination of microtubule binding and kinesin movement assays. Second there is the demonstration that KIF1C is important for the motility of Rab6 positive vesicle structures, and plays a role in the regulation of the structure of the Golgi apparatus.

These findings are combined in a model where Rab6 binding to two sites in KIF1C converts it from an active motor into a static tether.

*Reviewer 3*:

In this revision, the authors have focused on the concerns of reviewers 1 and 2 and strengthened the study with respect to some of those issues. The issues raised by reviewer 3 were not addressed. With respect to the first point, the authors still insist that loops 6 and 10 are responsible for interaction with Rab6, but the data simply is not there to demonstrate this. They should soften this statement. The second point relates to the functions of endogenous proteins, and in response the authors say they can show co-IPs of over-expressed proteins. The total lack of colocalization of Rab6 and KIF1C, especially in Rab6 over-expressing cells, is still troublesome to this reviewer, but not to the authors.

Overall, the in vitro interaction of Rab6-GTP with KIF1C is interesting, but I am not fully convinced that the in vivo function of this interaction as summarized in Figure 10 will stand the test of time.

[Editors' note: further revisions were requested prior to acceptance, as described below.]

Thank you for sending your work entitled “Rab6 regulation of the kinesin family KIF1C motor domain contributes to Golgi tethering” for further consideration at *eLife*. Your revised article has been favorably evaluated by Randy Schekman (Senior editor and Reviewing editor) and 3 reviewers.

The Reviewing editor and the reviewers discussed their comments before we reached this decision, and the Reviewing editor has assembled the following comments to help you prepare a revised submission.

You will see from the comments posted below that the three referees were mixed in their assessment of your responses to the concerns that were raised in the most recent submission of this work. Specifically, there is considerable doubt that you have or would be able to demonstrate that KIF1C is a Rab6-dependnet Golgi tether. Indeed, one referee went so far as to say that your data exclude this possibility. Another offered a balanced comment: “The first part of the study that Rab6 binds to KIF1C motor and tail domains seems robust, at least in vitro. The data indicating some role of KIF1C in Golgi organization is supported by previous data as well as new data in this study.” Another summarized the discussion with: “The biochemical analysis of the Rab6 interaction seems to be the higher quality part of the work. The imaging data relies on a rather odd method and I'm not sure I would draw any conclusions from it. As we discussed previously, it is obviously a concern that the authors cannot demonstrate localization of Rab6 and KIF1C. This leaves a number of obvious possibilities, two of which we have to acknowledge. First, that another Rab might be the major regulator of KIF1C in cells. Second, Rab6 binding is an in vitro pulldown artefact.”

After a lengthy consultation session we agreed that the only way we could consider this work for publication would be if you removed all mention of the Golgi tethering function of Rab6, and focused entirely on the strong evidence you have for Rab6 interaction with KIFC1. Your conclusions concerning the question of tethering will simply have to wait for more substantial and complete understanding of the physiological function of the Rab6-KIF1C-head interaction.

---

## [Author Response]

[Editors’ note: the author responses to the first round of peer review follows.]

Although the reviewers agreed that the finding was interesting and important, they felt that the story would be enhanced by: (A) determining the Kds for motor-microtubule interaction ± Rab6A protein; (B) and determining the Kd for Rab6A binding to the C-terminal Rab binding domain, to better understand any dominance that might exist between N- and C- terminal Rab binding sites; (C) testing if Rab6 prefers KIF1C-AMP-PNP over KIF1C-ADP. One reviewer noted a number of issues related to our in vitro motility assay. The purpose of the assay was to show a difference in microtubule occupancy with Rab6A (which is apparent), so we moved this to the supplemental material and discuss it in much less detail. Also, in some experiments it was difficult to show strong GTP preference. This turns out to be due to better nucleotide exchange with His-Rab6A compared with GST-Rab6A and we now see beautiful nucleotide specificity using GST-RabQL versus GST-RabTN mutant proteins (in some experiments we cannot use the His-Rab because the motor is His-tagged).

As summarized below, the new data reveal a greater than 10 fold effect of Rab6A on motor interaction with microtubules in both AMP-PNP and ADP; Rab6A prefers the AMP-PNP motor, and the motor binds Rab6A and microtubules with very similar affinity, setting up a wonderful target selection option—KIF1C can pick a microtubule or a Rab6-decorated membrane.

*The reviewers agree that the main finding that Rab6 interacts with both KIF1C motor and C-terminal domains and appears to regulate activity is interesting and important. However, the data in support of this conclusion are not strong. There are many clear controls that could have been done (e.g. the effect of Rab6 on the KIF-1C loop mutants interaction with MTs)*.

We thank the reviewers for taking time to evaluate the manuscript and provide constructive comments. We have added several key experiments that strengthen, significantly, our conclusions; we also have deemphasized some of the supplemental findings so as not to distract from our core message. Again, to our knowledge, no one has yet reported a protein that can bind to a motor domain and regulate its ability to bind microtubules. In addition, we show, for the first time, that a kinesin can function as a Golgi tether. Newly added Kd determinations reveal >10 fold effects (see below).

Reviewer 1:

*In this work, Lee et al. identify an interaction between KIF1C and the small GTPase Rab6a and suggest that this interaction serves to regulate the activity of this motor. In particular, they focus on a new interaction that they describe between the KIF1C motor domain and Rab6, documenting the region of interaction and effects on MT binding and motility. They then do in vivo experiments to show effects of KIF1C depletion or certain overexpression experiments on Golgi distribution and vesicle transport. Overall, there is a great deal of work contained in this paper (11 experimental main figures and supplement). However, I have reservations regarding the publication in* eLife*. First of all, the in vitro effects of the Rab6 on KIF1C microtubule interaction are relatively small (∼50%), which makes one wonder about the in vivo relevance. The GDP/GTP selection of Rab6 is also weak (50%;*
Figure 3*)*.

Rab6 does not release nucleotide well—unlike most other Rabs, crystallography has revealed that it retains bound, unhydrolyzed GTP even days after purification at 4°C. We realized that it is challenging to exchange GST-Rab6 with GDP while retaining full activity, *unlike His-Rab6A protein,* which exchanges more readily. To circumvent this we now show that the KIF1C motor domain strongly prefers the active GST-Rab6 QL mutant over the inactive GST-Rab6 TN mutant, with at least 8 fold preference (new Figure 3). Furthermore, *NEW Kd determinations reveal at least a 10 fold decrease in microtubule affinity.*

*Also, given that this is a core of the study, I thought that many of these in vitro experiments were not thoroughly documented as described below*.

We have better documented our experiments as requested.

*I also thought that in vivo experiments did not convincingly establish a role for the Rab6 nucleotide switch on regulating KIF1C motor/microtubule binding function in vivo*.

We show that Rab6 expression affects KIF1C interaction with microtubules in vivo and this is dependent on loops shown to be required for Rab6 binding. Rab6 nucleotide specificity in vivo is less reliable because overexpressed, mutant Rab6 TN can still interact with KIF1C (at lower levels compared to Rab6 QL), and controlling the level of expression between overexpressed constructs is difficult to achieve.

*Thus, while the data is interesting individually, I did not find that the paper as a whole conveyed a compelling story, or at least at the level required for an* eLife *publication. A major defect in the paper was the characterization of in vitro effect of Rab6 on KIF1C*.

This has been remedied.

Figures 6 and 7
*(the core of the paper for showing effects on microtubules) should be done with the same protein, and ideally without a KIFC-KIF5 fusion. Ideally, the authors should try baculovirus expression to have a well-behaved protein that can be used in both sets of experiments*.

Previous Figure 6 (new Figure 5) shows cosedimentation in which Rab6 affects *IVT produced, full length KIF1C’s* ability to bind to MT. In previous Figure 7 (new Figure 5—figure supplement 1), which shows similarly that Rab6 regulates KIF1C’s MT interaction by microscopy, we did use *Baculovirus to produce full length KIF1C* that was able to processively walk on MTs. In other figures, it was critical for us to be able to test just the motor domain to show that Rab6 regulation of KIF1C’s MT binding is localized to the motor domain alone. Please keep in mind that the “fusion” represents 349 residues of KIF1C plus only 6 residues of KIF5—these 6 yield a protein that can bind 100% to microtubules in vitro, unlike the IVT-produced 1-350 KIF1C motor that is only ∼3% active by this criterion. Only with this form of the motor are we able to carry out reliable Kd determinations for MTs shown now in the new Figure 5.

*a) The microtubule binding experiments should be done by performing a microtubule binding curve to measure Kd and should be done with control, Rab6-GDP and Rab6GTP*.

We thank the referee for this particularly helpful suggestion. We have now measured a Kd for KIF1C motor domain binding to MTs in the presence of control or active Rab6 QL and we show now that Rab6 binds with preference to the motor with AMP-PNP bound (NEW Figure 3). We also carried out MT affinity experiments and report that in AMP-PNP, KIF1C motor binds MTs with Kds of 0.46µM versus ∼5µM in the presence of Rab6. Similarly, in ADP, KIF1C motor binds MTs with Kds of 2µM versus >15µM in the presence of Rab6 (New Figure 5). These new data provide strong additional support for the conclusions of this paper.

The reviewer asked for comparison of GTP and GDP forms. This was not possible because the GDP needed to stabilize the Rab changed the behavior of taxol stabilized microtubules, decreasing the efficiency of MT polymerization/stability/sedimentation and making it difficult to distinguish if any KIF1C was interacting with non-sedimenting GDP-tubulin. Instead we include a Rab33 control (Figure 5).

*b) The*
Figure 7
*motility experiments are also suboptimal. First of all, they were done with an antibody and secondary antibody, which can result in crosslinking, and this is just very complicated to interpret. Direct labeling (e.g SNAP or Halo) would be better*.

Yes, but we were only trying to show that Rab6 would affect the ability of KIF1C to stay on MTs, which it does. We have moved this figure to supplemental as it represents corroboratory rather than essential data.

*Second, there is no measurement of velocity or processivity*.

These were measured but not introduced as no significant difference was seen in the presence or absence of KIF1C; we state this in the text.

*Third, Rab6 should be tested for effects again in the GDP and GTP states*.

This would be difficult, as nucleotide changed MT polymerization (see above).

*Fourth, the very low number of moving motors is troublesome; suggested to be due to autoinhbition but not really followed up. Fifth, it would be good to establish whether this KIF1C construct is monomeric or dimeric (or contains aggregates). Sixth, would be good to label Rab6 to see if it is co-bound to the motors on the microtubule (or not)*.

All we wish to conclude is that the Rab changes MT association. The observations from Figure 7 that Rab6 affects KIF1Cs ability to bind MTs confirm our co-sedimentation findings, which we have strengthened substantially; again, we now present Figure 7 as Figure 6*–*figure supplement 1.

Reviewer 2:

*In this study Pfeffer and coworkers describe and characterize the interaction of Rab6 with KIF1C, and describe a role in Golgi-associated transport. The demonstration that Rab6 can directly interact with and regulate the kinesin motor activity is an important finding that would justify publication in* eLife*. I have a few major comments*.

*1) The authors provide clear evidence for the interaction of Rab6a with the c-terminal region of KIF1C. This is highly specific for the GTP-bound form consistent with the proposal that KIF1C is a Rab6 effector. Surprisingly deletion of this region did not abolish Rab6 binding and the authors identify a second binding site in the n-terminal motor region. There is preferential binding to Rab6-GTP but not to the same extent as the CBD site. It would be interesting to know how the CBD affinity compares to the motor site ∼0.2µM. Does the Rab6 GTP specificity data indicate that CBD binds more tightly? The relationship between the two sites is important for understanding how Rab6 is recruited. If one site has a much higher affinity than the other then one would expect it to be occupied first, having important implications for the system*.

Please see above regarding the GTP/GDP preference of the motor domain. As requested, we now include the Kd for Rab6A to the motor and binding is tight—0.9µM (new Figure 1). This indicates that binding is comparable to both N- and C-terminal sites. In cells, the C-terminus may alternately be affected by bicaudal D1, thus one cannot make too many conclusions regarding the two ends of KIF1C— although it is unlikely that one binding site predominates in the interaction. Interestingly, the Kds for Rab6A and microtubules are also very similar (0.2 versus 0.46µM).

*2) By preventing motor activity with the K103A mutation, KIF1C is converted from a dynamic motor/tether to a static tether. When overexpressed in cells this construct may suppress Golgi ribbon defects, but this is not the same as proving motor activity is not normally required for KIF1C function in the system*.

We apologize for the confusion, but are definitely not meaning to say that motor activity is not required for total overall function, only its function as a Golgi tether. We tried to clarify the text

*For example in the speed and direction change assays in*
Figure 9
*one would imagine the motor inactive KIF1C fails to rescue these aspects of Rab6 vesicle movement. The K103A rescue also looks slightly different to the wild-type KIF1C condition in the Golgi ribbon assay, so it clearly does not rescue completely. This possibility should be mentioned since it is relevant for the discussion of whether the motor domain interaction is a way of creating a Golgi ribbon tether or perhaps a form of motor activity regulation*.

K103A mutant really seems to be pretty normal in the rescue assay—please compare the images in new Figures 8 and 9 (old figure 10 and 11) and their supplements as well as the quantitation of the rescue phenotypes. There was no significant difference between the wild-type rescue and the K103A rescue. The inability of this motor to move is shown in Figure 8—figure supplement 2 where it seems to sit perinuclearly. We have tried to clarify the text as requested.

*I wasn't entirely sure of the argument being made in the subsection headed “Rab6A-KIF1C interaction is required for maintenance of normal Golgi structure” regarding the wild type and K103A mutant. If wild-type KIF1C fails to rescue, then surely the fact a mutant does rescue implies there is an important difference between the two relevant for normal function*.

We apologize for the confusion.

The argument is that wt KIF1C along with K103A mutant both rescue the siRNA depletion. We have tried to make the text clearer.

*3) Does the ADP/ATP state of the motor domain make any difference to the Rab6A binding shown in*
Figure 5*? This is relevant for understanding the relationship between kinesin-microtubule interaction and Rab6A binding and the data in*
Figures 6 and 7.

Yes. We now show that Rab6A binds to KIF1C more strongly in the AMP-PNP state over the ADP state (See new Figure 3).

*4) Do the KIF1C loop mutants have microtubule-stimulated ATPase activity? This should be shown or mentioned if it has been tested*.

They are capable of binding MTs as shown in Figure 4—figure supplement 2 and old Figure 8 (new Figure 6); we have not prepared the loop mutants in amounts that would permit this assay to be carried out.

Reviewer 3:

*There seem to be a couple of issues: 1) A key result involves switching loops 6 and 10 out of KIF1C and replacing them with the corresponding ones from KIF1A. This negative result is interpreted as showing that the loops bind Rab6. There are other possible interpretations, most notably that the conformation is disturbed. Obviously, if the converse switch of loops 6 and 10 from KIF1C into KIF1A induces Rab6 binding, this would be excellent evidence. Failing this, the docked structure shown in*
Figure 5
*should reveal critical residues in the loops that could be mutated to make a stronger case. The switched loop construct is used throughout the study—if loops 6 and 10 cannot be shown to be directly involved in Rab6 binding, this does not take away from the rest of the story, as this construct had a clear defect in Rab binding*.

Given the work required to generate all of the additional new data, we have not had time to make the opposite mutant but agree it would be interesting although as the reviewer has noted, not necessary to the overall conclusions of the paper.

*2) The data are consistent with an interaction between Rab6 and KIF1C based on coIP of over-expressed proteins, and in vitro studies. This is perhaps a bit surprising as the images in*
Figure 8
*and*
Figure 9
*suggest that over-expressed KIF1C is almost completely peripheral, whereas over-expressed Rab6 is almost all perinuclear. Can the endogenous proteins be seen to coIP or colocalize? Wouldn't a consequence of their model suggest that over-expression of Rab6 should drive KIF1C to the Golgi complex? This issue brings into question whether Rab6 is in fact the correct Rab protein that endogeneous KIF1C interacts with*.

KIF1C’s propensity to move to the tips of cells has been challenging for us—Golgi Rab6 is occupied with other partners. Nevertheless, most KIF1C is functioning as an anterograde motor, not a tether. Because of this, we do not expect the motor to sit quietly at the Golgi, however the Golgi depletion phenotype is robust. We do show co-IP from cells in Figure 1; KIF1C also influences the motility of vesicles carrying GFP-Rab6 to the periphery (New Figure 7 (old Figure 9). But we don’t have any antibody that recognizes the endogenous KIF1C protein by IF (only western blot). Lammers et al. reported a Golgi stain and we can detect KIF1C in rat liver Golgi fractions (see Figure 10). Our confidence that Rab6 is key comes from the rather high affinities that this Rab shows for KIF1C, relative to the affinities of most Rab effectors, which are mostly in the low micromolar (vs. ours in the high nanomolar) range.

Author response image 1.**DOI:**
http://dx.doi.org/10.7554/eLife.06029.023

[Editors' note: the author responses to the re-review follow.]

*There continues to be a considerable degree of concern that you have not been able to colocalize KIf1C and Rab6A on Golgi membranes, and that without such a demonstration, the claim that KIF1C serves as a Golgi tether is in doubt. In the consultation session, there was discussion about further attempts to document this colocalization with other antibodies, but there was some feeling that this has not been successful in other laboratories and may not be feasible at this time. Under the circumstances, the view is that you should alter your title and conclusions to remove the tether assignment for KIF1C. Such a modification would be acceptable to the referees. However, you may wish to make one more attempt at colocalization, which if successful would considerably improve the impact of your work. Let us know how you wish to proceed*.

We have now been able to detect KIF1C in contact with the Golgi using a permeabilization approach described by Mathew Seaman in 2004. This method uses liquid nitrogen-cell breakage to leach cytosol from cells prior to fixation. We observe tagged KIF1C in contact with Golgi membranes, with either wild type or K103A KIF1C, with or without Rab6A co-expression, with or without nocodazole (new Figure 9 and Figure 9–figure supplement 2). Even though some membrane-associated proteins in equilibrium with cytosol may be lost, we do detect KIF1C contacts with Golgi membranes in all of these conditions. In addition, we have gone through the text and carefully modified our conclusions throughout to say that the protein “contributes to” tethering and we have also changed the title so as never to state that it is a bona fide tether. We believe our tethering phenotype and KIF1C rescue data are highly compelling to membrane traffickers (Figure 8 and Figure 9) and hope that the reviewers agree that the new micrographs support our overall conclusions.

Reviewer 1:

*This paper describes the interaction between kinesin-3 KIF1C and the small GTPase Rab6. The authors show that Rab6A can specifically interact both with the motor and the tail domain of KIF1C. The authors suggest that the function of this interaction is two-fold: to mediate transport of Rab6A vesicles to the cell periphery, and to act as a Rab6A-specific inter-membrane Golgi tether*.

*Participation of KIF1C in the microtubule plus end-directed transport of Rab6A-decorated vesicles has already been documented (Schlager et al., EMBO J, 2010; Schlager et al., Cell Reports 8(5):1248-56, 2014; note that the second paper is recent and is not cited). The finding that a domain in KIF1C tail binds to Rab6A certainly helps to understand how the motor is recruited to the vesicles, but by itself does not seem to justify publication in* eLife*.*

We thank the reviewer for bringing the additional citation to our attention and have added it here. In the second paper, Schlager et al. artificially recruit KIF1C motor domain to Rab6 vesicles and their results differ from ours. When they do this, they see velocity of Rab6 vesicles increase whereas we saw an increase in speed following KIF1C depletion. Our study reflects native motors, not those forced onto Rab6 vesicles—the native Rab6A vesicles are likely decorated with other proteins that may not be on the vesicles analyzed in the Schlager story.

*A more surprising claim in this study is that Rab6A can also bind to the KIF1C motor domain, prevent its interaction with microtubules and convert it into an inter-membrane tether. The biochemical data in the revised paper seem mostly reasonable. However, I have strong reservations about the cell biological data. If KIF1C serves as a Rab6A-dependent, functionally relevant Golgi tether—a protein that can connect vesicles to Golgi stacks or promote association between different Golgi stacks, like the authors suggest—this kinesin must be detectable at the Golgi. However, the paper does not contain a single image showing the Golgi localization of KIF1C, either endogenous or overexpressed. For example, in*
Figure 9*, the authors show data indicating that overexpressed KIF1C or KIF1C 103A mutant suppress nocodazole-mediated Golgi dispersion. It would seem that KIF1C should be detectable at the Golgi in these conditions, especially as microtubules are not present and should not compete with Rab6A for binding. In the rebuttal letter, the authors indicate that there are no antibodies that can detect endogenous KIF1C. This statement does not seem accurate, because two papers (Theisen et al., Dev Cell 2012, Efimova et al., J Cell Sci 2014) successfully used KIF1C antibodies from Cytoskeleton to stain endogenous KIF1C (although no specific accumulation of KIF1C at the Golgi seemed to be present in the images in these papers). As an alternative to endogenous staining, a better effort could be made with tagged KIF1C and tag-specific antibodies. Ideally, the authors should show that there is a KIF1C pool at the Golgi, and that this pool is lost when Rab6A is depleted. If no KIF1C can be detected at the Golgi, then the most trivial explanation is that its levels at this membrane organelle are very low, bringing the* “*tether*” *model into question*.

*Other comments*:

Figure 5—figure supplement 1*: The data do not look convincing, the percentage of moving motors is extremely low, which in this type of assay typically indicates misfolding of the kinesin motor domain. Also the effect of more than a 1000-fold (!) excess of Rab6A on microtubule binding is weak*.

The experiment in Figure 5—figure supplement 1 was done with full length, Baculovirus produced KIF1C, and we agree that it is not fully active. Relevant here is the concentration in relation to the Kd, not the fold excess. We are in <30 fold excess (above the Kd) for each binding site on KIF1C. All we conclude here is that Rab6 added above the Kd interferes with binding—which matches the microtubule sedimentation shown in Figure 5. Experiments with the full active motor domain presented in Figure 5 shows the ability of Rab6A to interfere with motordomain microtubule binding as a function of Rab6A concentration and binding inhibition is seen at much lower Rab6A concentrations than happen to have been used in Figure 5—figure supplement 1. The results are entirely consistent and were obtained using complementary assays. We prefer to keep this figure supplement in the paper for this reason.

Figure 6*: Overexpressed Rab6A should be mostly present on the Golgi and vesicles. If the affinity between Rab6A and KIF1C is sufficiently high to* “*pull the motor off*” *the microtubules, the motor should relocalize to the Rab6A-positive membranes, an effect that should be detectable in PFA-fixed cells*.

We show that KIF1C comes off microtubule tracks upon Rab6 overexpression. Some KIF1C does appear to colocalize with Rab6A vesicles in both MeOH and PFA treated cells but is difficult to quantify. In PFA treated cells, both Rab6A and KIF1C occupy a large cytosolic pool in addition to their vesicle localization, which makes analysis challenging. For MeOH treated cells, much of the non-microtubule associated KIF1C pool and Rab6 have been washed away. While Rab6 correlates with microtubules and KIF1C motor domain also localizes to microtubules, it is difficult to distinguish what is colocalized versus what is coincidentally interacting with the cytoskeleton. Importantly, we have now included full length KIF1C localization under liquid nitrogen treatment, which shows numerous contacts with the Golgi and Rab6A (see above).

Figures 6 and 7*: The effect of KIF1C depletion on the vesicle movement is mild, in agreement with published data, and is likely explained by participation of additional kinesins such as KIF5B. It is therefore unclear why the effect on VSV-G secretion is so strong. The authors say that rescues were not possible. Can they exclude that it is an off-target effect of the siRNA (for example, by showing knockdown with another siRNA)? Using at least two different siRNAs to show a phenotype seems now standard in the field*.

Simpson et al. independently reported that KIF1C depletion resulted in inhibition of VSV-G transport using a different KIF1C siRNA. While the full mechanism of KIF1C depletion on VSV-G secretion is still undetermined, the speed of vesicle movement may play a minor roll, with changes in processivity, Golgi structure, and containment possibly contributing to VSV-G export efficiency. Again this is a very minor point and not the main conclusion here.

Reviewer 2:

*In this study Pfeffer and coworkers show that the kinesin KIF1C is regulated by the small GTPase Rab6. I felt the original submission was an important piece of work that could have been published in* eLife*. There were some issues that needed to be addressed and most of these have been dealt with in the revised submission*.

*I think it is important to clarify what the authors have found. First there is the identification and characterisation of two Rab6 binding sites in KIF1C. One of these sites lies within the kinesin motor domain. The authors provide evidence that this site is important for regulating KIF1C using a combination of microtubule binding and kinesin movement assays. Second there is the demonstration that KIF1C is important for the motility of Rab6 positive vesicle structures, and plays a role in the regulation of the structure of the Golgi apparatus*.

*These findings are combined in a model where Rab6 binding to two sites in KIF1C converts it from an active motor into a static tether*.

We thank the reviewer for appreciating what this paper shows.

Reviewer 3:

*In this revision, the authors have focused on the concerns of reviewers 1 and 2 and strengthened the study with respect to some of those issues. The issues raised by reviewer 3 were not addressed. With respect to the first point, the authors still insist that loops 6 and 10 are responsible for interaction with Rab6, but the data simply is not there to demonstrate this. They should soften this statement*.

We have softened our language to state that loops 6 and 10 are required for Rab6 interaction with the motor domain rather than being “responsible” for the interaction.

*The second point relates to the functions of endogenous proteins, and in response the authors say they can show co-IPs of over-expressed proteins. The total lack of colocalization of Rab6 and KIF1C, especially in Rab6 over-expressing cells, is still troublesome to this reviewer, but not to the authors*.

*Overall, the in vitro interaction of Rab6-GTP with KIF1C is interesting, but I am not fully convinced that the in vivo function of this interaction as summarized in Figure 10 will stand the test of time*.

Please see the comments above where we can now see KIF1C in the vicinity of the Golgi using liquid nitrogen fixation. We have also softened our conclusions as not to mislead the reader or overstate our findings.

[Editors' note: further revisions were requested prior to acceptance, as described below.]

*[…] After a lengthy consultation session we agreed that the only way we could consider this work for publication would be if you removed all mention of the Golgi tethering function of Rab6, and focused entirely on the strong evidence you have for Rab6 interaction with KIFC1. Your conclusions concerning the question of tethering will simply have to wait for more substantial and complete understanding of the physiological function of the Rab6-KIF1C-head interaction*.

As stipulated in your decision letter, the text has been carefully edited to remove *any* mention of the possible Golgi tethering function of KIF1C; instead it focuses entirely on our evidence that Rab6 interacts with KIFC1, and KIF1C is needed for proper Golgi organization. We have also removed a summary-model figure to be sure that there is no confusion in terms of what we conclude here.

We have incorporated data from a supplemental figure into the main panel of Figure 9 showing contacts of KIF1C with Golgi membranes, and have adjusted the contrast, as the co-localization may not have reproduced well in the journal-generated PDF image. Six different cells are shown as examples, and we don’t make any strong conclusions about what is shown there (we disagree with the reviewer who concluded that no contacts were seen; see Figure 9). In any case, this helps reduce the number of supplemental figures and hopefully clarifies the presentation. (A referee had asked for a loop mutant KIF1C but such a protein can still bind Rab6 so would not be a good control.)